# Individual bat virome analysis reveals co-infection and spillover among bats and virus zoonotic potential

Jing Wang[1,2,12], Yuan-fei Pan [3,12], Li-fen Yang[4,12], Wei-hong Yang[4], Kexin Lv[5], Chu-ming Luo[5], Juan Wang[4], Guo-peng Kuang[4], Wei-chen Wu[1,2], Qin-yu Gou[1,2], Gen-yang Xin[1,2], Bo Li [6], Huan-le Luo [5], Shoudeng Chen [7], Yue-long Shu[5], Deyin Guo [1,8], Zi-Hou Gao[4], Guodong Liang[9], Jun Li [10], Yao-qing Chen [5] ✉, Edward C. Holmes [11] ✉, Yun Feng[4] ✉ & Mang Shi [1,2] ✉

Bats are reservoir hosts for many zoonotic viruses. Despite this, relatively little is known about the diversity and abundance of viruses within individual bats, and hence the frequency of virus co-infection and spillover among them. We characterize the mammal-associated viruses in 149 individual bats sampled from Yunnan province, China, using an unbiased meta-transcriptomics approach. This reveals a high frequency of virus co-infection (simultaneous infection of bat individuals by multiple viral species) and spillover among the animals studied, which may in turn facilitate virus recombination and reassortment. Of note, we identify five viral species that are likely to be pathogenic to humans or livestock, based on phylogenetic relatedness to known pathogens or in vitro receptor binding assays. This includes a novel recombinant SARS-like coronavirus that is closely related to both SARS-CoV and SARS-CoV-2. In vitro assays indicate that this recombinant virus can utilize the human ACE2 receptor such that it is likely to be of increased emergence risk. Our study highlights the common occurrence of co-infection and spillover of bat viruses and their implications for virus emergence.

Bats (order Chiroptera) are hosts for a larger number of virus species than most mammalian orders[1], and are the natural reservoirs for several emerging viruses that cause infectious disease in human[2]. Recently, there has been considerable research effort directed toward exploring viral diversity in bats as a means to identifying potential zoonotic infections[3]. These studies have greatly expanded the diversity of known bat-borne viruses and identified an array of potentially emerging viruses. However, despite the growing body of work on bat viruses, little is known about the underlying drivers of virus diversity within these animals, nor of the extent and patterns of viral co-infection and the frequency of viral spillover among bat species[4,5].

Current virus discovery studies typically pool individual bats by species or by sampling location[6,7]. Although of great utility, this hinders mechanistic insights due to insufficient resolution. As such, studying the bat virome at the scale of individual animals can help us better understand the diversity and emergence of bat-borne viruses[4]. For example, the co-infection of phylogenetically related viruses within an individual host facilitates the occurrence of recombination or reassortment in the case of segmented viruses[8], which may in turn have contributed to the emergence of a number of zoonotic viruses (e.g., SARS-CoV[9]). Importantly, the frequency of virus co-infection in bats can be resolved through the study of the viromes of individual animals. Resolution at the scale of individual animals is also required to better understand the frequency and determinants of virus spillover among bats[4,10], and to reduce the impact of potential confounding effects.

A full list of affiliations appears at the end of the paper. ✉e-mail: chenyaoqing@mail.sysu.edu.cn; edward.holmes@sydney.edu.au; ynfy428@163.com; shim23@mail.sysu.edu.cn

Many previous studies of bat viruses have preferentially targeted relatives of known human pathogens[3]. Although time and cost-effective, this necessarily limits our ability to discover novel zoonotic viruses. In contrast, other studies have utilized metagenomics approaches to explore the total bat virome[7], with meta-transcriptomic sequencing demonstrating great utility as a means to characterize the total diversity of viruses without a priori knowledge of which viruses are present[11,12].

Yunnan province in southwestern China has been identified as a hotspot for the diversity of bat species and bat-borne viruses. A number of potential zoonotic viruses have been detected there, including close relatives of SARS-CoV-2, such as bat coronavirus RaTG13[13], RpYN06[14], and RmYN02[15], as well as relatives of SARS-CoV, such as WIV1[16] and Rs4231[17]. It has been hypothesized that the presence of mixed roosts of bats in Yunnan (i.e., multiple bat species occupying the same roost) contributes to the frequent cross-species transmission of viruses, promoting their recombination and ultimately leading to transient spillovers or successful cross-species transmissions[17]. Thus, wild bat populations in Yunnan provide a unique opportunity to study the diversity, spillover and emergence risk of bat-borne viruses.

We performed intensive field sampling of individual wild bats in Yunnan province, China. In particular, we characterized the total mammal-associated virome of wild bats (i.e., viruses that are likely to infect bats based on their phylogenetic relatedness to known viruses of mammalian hosts, in contrast to viruses associated with the bat microbiome or diet) at the scale of individual animals using unbiased meta-transcriptomic sequencing. We then explored the cross-species transmission of viruses among individual animals from different species and quantitatively tested how host phylogeny and geographic (i.e., sampling) location may impact the probability of cross-species transmission. Finally, we identified viruses of potentially high emergence risk and evaluated their pathogenic potential using a combination of phylogenetic analysis, and in silico simulations as well as in vitro receptor binding affinity experiments.

## Results

### Characterization of the bat viromes

Between 2015 and 2019, rectum samples were collected from 149 individual bats located in six counties/cities, Yunnan province, China. These represented six bat genera and 15 species (Fig. 1; Supplementary Fig. 1 and Supplementary Table 1). Total RNA was extracted and sequenced separately for each individual bat. Meta-transcriptomic sequencing yielded an average of 41,789,834 clean non-rRNA reads for each animal, from which 1,048,576 contigs were de novo assembled. We searched for assembled contigs that encode hallmark genes of viruses (e.g., the RNA-dependent RNA replicase (RdRp) for RNA viruses), from which 758 viral contigs were identified after filtering for contig length and hallmark gene completeness. A taxonomic placement was then assigned to each viral genome according to both the amino acid sequence identity of hallmark genes and nucleotide sequence identity at the whole genome scale based on the ICTV (International Committee on Taxonomy of Viruses) criteria.

We then focused our analysis on characterizing the mammal-associated viromes of bats (Fig. 2), which represent those RNA and DNA viral families or genera that are known to infect mammalian hosts (rather than those viruses more likely associated with bat diet or microbiome). Accordingly, we identified 55 mammal-associated virus species belonging to 12 families (Supplementary Table 2 and Supplementary Figs. 2 and 3). Most of the viruses detected were RNA viruses, comprising 46 of the 55 viral species. The *Reoviridae* was the most prevalent viral family, present in 27.5% of individuals sampled, followed by the *Picornaviridae* (16.1%) and the *Coronaviridae* (9.4%) (Fig. 3c). The prevalence of the remaining viral families was relatively low (≤6%).

We next quantified viral abundance and the number of virus species for each individual bat (Fig. 3). Of the 149 individual bats analyzed, 73 were positive for at least one virus species (positive rate 49%). We consider those viruses with relatively high abundance (reads per million total reads >1) as positives, and we used RT-PCR to confirm the existence of a subset of viruses with low read coverage (<30%). These criteria were previously shown to result in a low false-positive

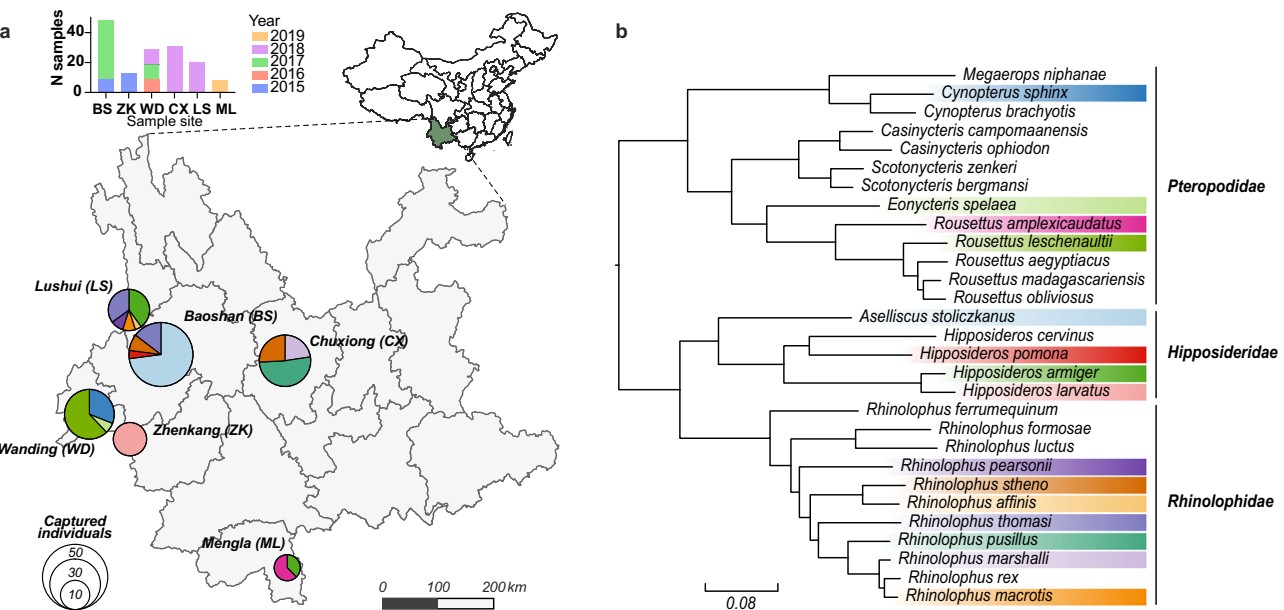

**Fig. 1 | Overview of the samples analyzed in this study. a** Locations in Yunnan province China where bat samples were taken. Bar plot on the top shows number of samples per year per site. Pie charts indicate the composition of bat species sampled at each location, while the total area of the pies are proportional to number of captured individuals. Colors indicate different bat species, which are consistent with the coloring scheme in plot (**b**). **b** Phylogeny of bats, including those sampled as part of this study. The tree was estimated using nucleotide sequences of bat COI gene utilizing a maximum likelihood (ML) method. Colored strips indicate the bat species sampled in this study. Map data were retrieved from 10.5281/zenodo.4167299. Source data are provided as a Source Data file.

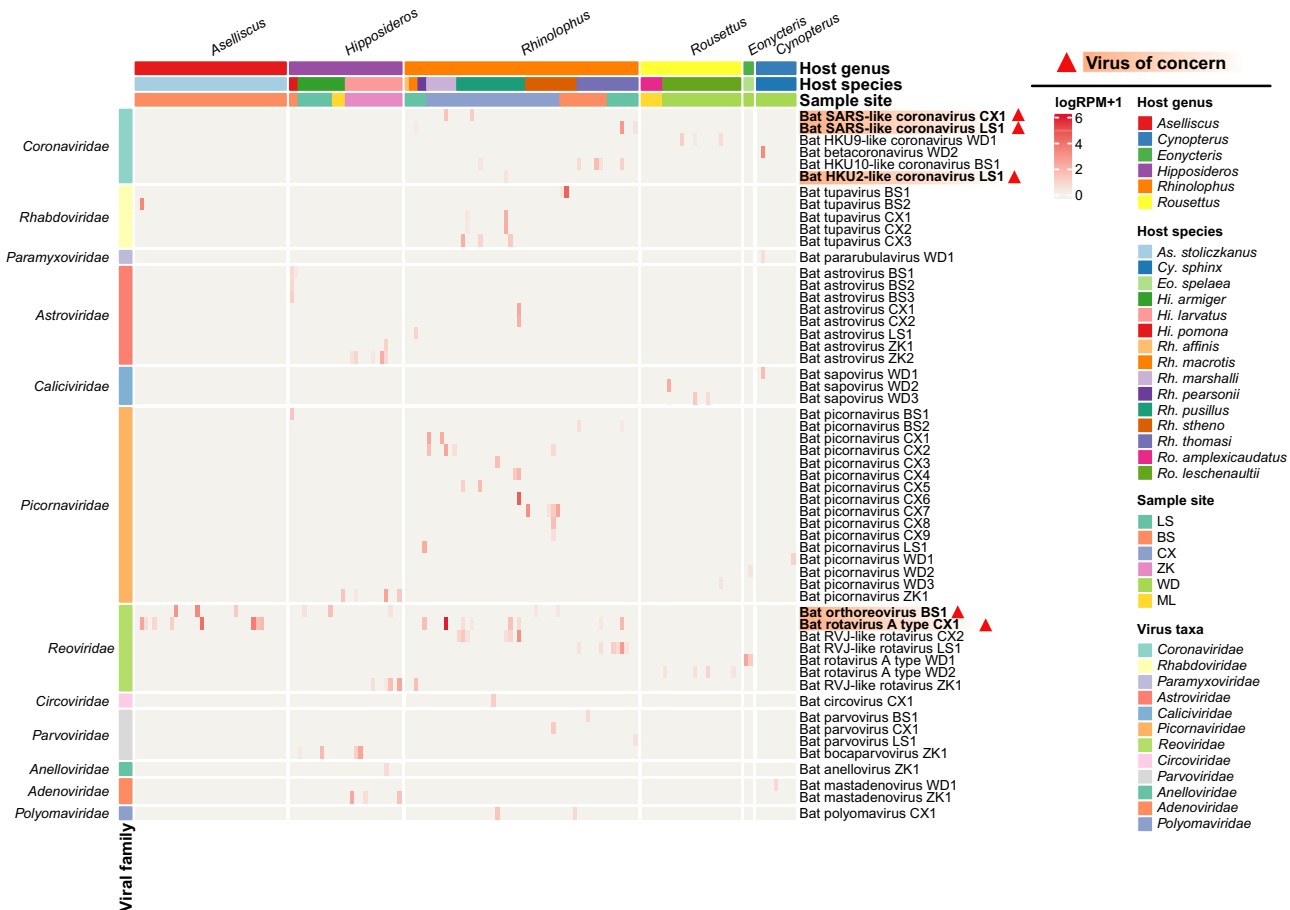

**Fig. 2 | Characterization of the mammal-associated virome of bats.** The heatmap displays the distribution and abundance of mammal-associated viruses in individual bats. Each column represents an individual bat, while each row represents a virus species. The abundance of viruses in each individual is represented as a logarithm of the number of mapped reads per million total reads (RPM). Sampling site, host taxonomy (species and genus) and virus taxonomy are shown as colored strips at top and left, respectively. Red triangle marks indicate "viruses of concern", defined as those that are closely related to known human or livestock pathogens (>90% amino acid similarity in RNA-dependent RNA polymerase). Source data are provided as a Source Data file.

rate[18]. Among virus-positive individuals, 42% were infected by more than one viral species (with an average of 1.8 viral species per individual). The number of virus species per individual was uneven among host genera (Fig. 3b).

**Cross-species transmission of viruses among bats**
After applying multiple false-positive controls, we identified ten virus species that were shared among different bat species (Fig. 4a), belonging to the *Coronaviridae* (3 species), *Reoviridae* (3), *Picornaviridae* (2), *Parvoviridae* (1) and *Polyomaviridae* (1). The whole genome nucleotide identity of the sequences from each viral species in different bat species ranged from 78% to 100%, with most having nucleotide identities >90% (Supplementary Fig. 4). Bat rotavirus A type CX1 and Bat orthoreovirus BS1 were detected in more host species than the other viruses, being detected in five and four bat species, respectively. The remaining nine viral species were only found in two bat species, and most were only shared among animals from the same host genus, with the exception of Bat RVJ-like rotavirus ZK1, which was present in *Hi. larvatus* and *Rh. macrotis*.

An analysis using partial Mantel tests revealed that more closely phylogenetically related or closely geographically located bat individuals had more similar mammal-associated virome compositions and had more virus species in common (Supplementary Table 4). A parallel analysis for total virome compositions (including all viruses, such as bacteriophage) provided similar results (Supplementary Table 5 and Supplementary Figs. 5–7). For example, the viromes of *Rhinolophus* or

*Hipposideros* bats form two network modules, in which individuals within the same genus are more inter-connected (i.e., shared more viruses) than individuals from different genera (Fig. 4a). We further used Poisson regression analysis to show that the number of virus species shared between pairs of individual bats was significantly associated with both the phylogenetic and the geographic distance of hosts, after controlling for the confounding effect of sampling date (Fig. 4b, c).

**Identifying viruses of potentially high emergence risk**
Phylogenetic analysis identified five viral species that were closely related to known human or livestock pathogens, which we denoted "viruses of concern" (Fig. 5 and Tables 1 and S5). The five viruses of concern belonged to two viral families—the *Coronaviridae* (three species) and the *Reoviridae* (two species). Notably, four of the five viruses (excluding Bat HKU2-like coronavirus LS1) were detected in more than one bat species; for example, Bat SARS-like coronavirus CX1 was detected in *Rh. pusillus* and *Rh. marshalli*, while Bat SARS-like coronavirus LS1 was detected in *Rh. macrotis* and *Rh. thomasi*. The prevalence of these viruses of concern was relatively high, especially Bat orthoreovirus BS1 and Bat rotavirus A type CX1 (Table 1).

The three coronaviruses were closely related to known zoonotic viruses that infect humans or swine. A phylogenetic analysis using the RdRp protein revealed that both Bat SARS-like coronavirus CX1 and LS1 belonged to the subgenus *Sarbecovirus* of *Betacoronavirus* and are closely related to human SARS-CoV (>90% nucleotide identity).

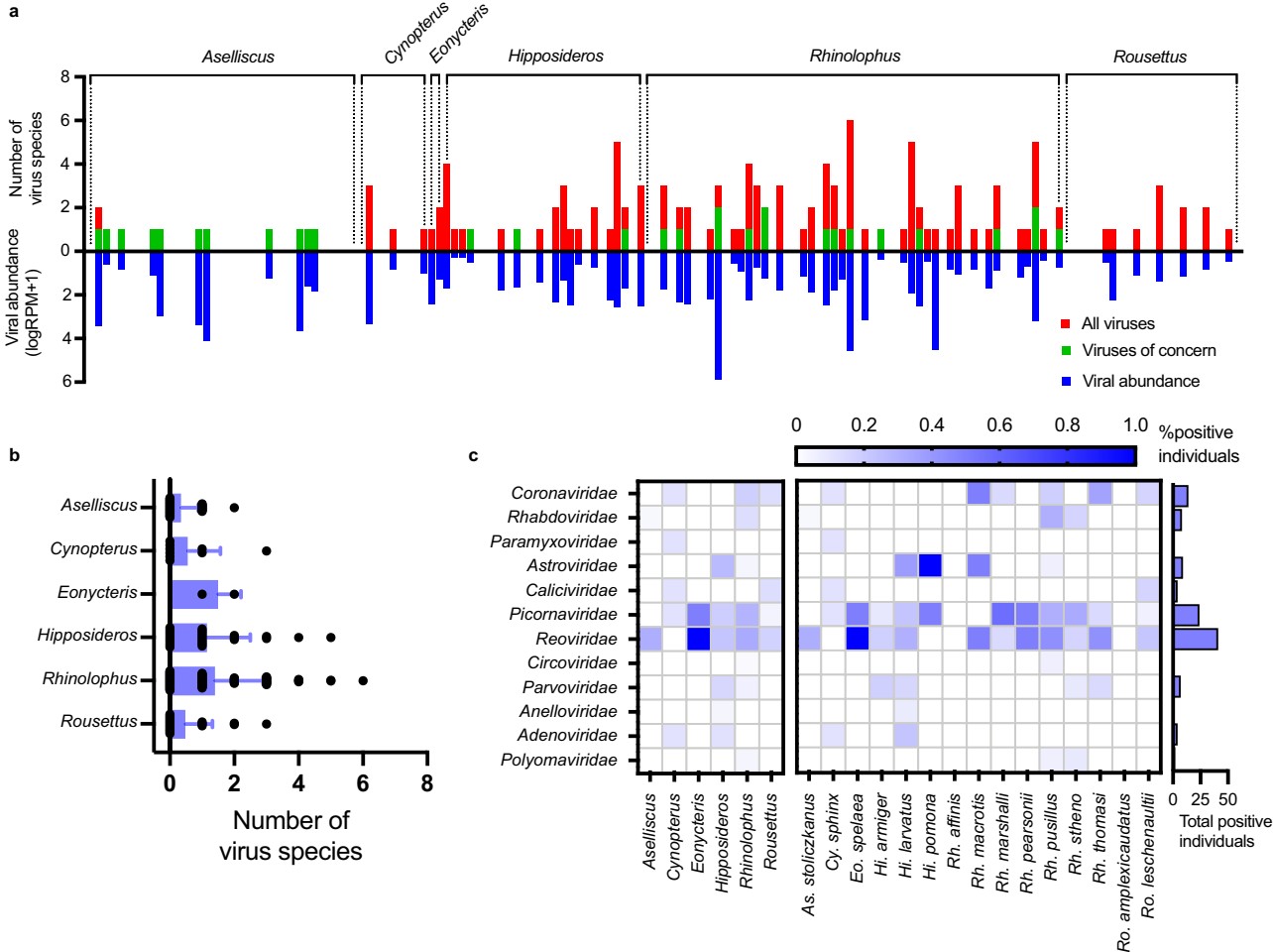

**Fig. 3 | Comparison of mammal-associated virus diversity among different bat taxa. a** Virus abundance and the number of virus species in individual bats. Red bars, the total number of mammal-associated viruses per host. Green bars, number of viruses of concern per host. Blue bars represent viral abundance per host as logarithm of the sum of total viral RPM. **b** Comparison of the number of viruses per individual host among six bat genera (mean + SD). Sample size: *Aselliscus n* = 35 individual bats, *Cynopterus n* = 9, *Eonycteris n* = 2, *Hipposideros n* = 26, *Rhinolophus n* = 72, *Rousettus n* = 23. **c** Comparison of the prevalence of 11 viral families among different host genera (left block) and species (right block). Source data are provided as a Source Data file.

Notably, other key functional genes or domains (e.g., NTD, RBD, N) of Bat SARS-like coronavirus CX1 were more closely related to SARS-CoV-2 (i.e., the early Wuhan-Hu-1 reference strain), indicative of a past history of recombination. We present further analysis of the evolutionary history and zoonotic potential of this virus below. The other coronavirus—Bat HKU2-like coronavirus LS1—belonged to the genus *Alphacoronavirus* and was closely related to Swine acute diarrhea syndrome coronavirus (SADS-CoV) in the RdRp gene (93.6% nucleotide identity).

The remaining two viruses of concern, Bat orthoreovirus BS1 and Bat rotavirus A type CX1 from the *Reoviridae*, were closely related to Mammalian orthoreovirus and human-infecting variants of Rotavirus A respectively. Indeed, we identified three rotaviruses (Bat rotavirus A type CX1, WD1 and WD2) that were different genotypes of the same viral species, Rotavirus A, based on the nucleotide similarity of all the 11 genome segments. Bat rotavirus A type WD1 and WD2 were more distantly related to human-infecting variants of Rotavirus A (RdRp protein <90% amino acid identity) than CX1. Furthermore, the observed host range of the three genotypes were distinct. Bat rotavirus A type CX1 was detected in *As. Stoliczkanus* and multiple *Rhinolophus* species, while Bat rotavirus A type WD1 and WD2 were associated with *Eonycteris* and *Rousettus* bats respectively (Supplementary Fig. 2).

As well as viruses of concern, 42 viral species were classed as newly discovered viruses (hallmark genes <90% amino acid identity, or whole genome <80% nucleotide identity to any existing viral sequences). The *Picornaviridae* (n = 14) contained the highest number of the newly discovered viral species, followed by the *Astroviridae* (n = 8), *Rhabdoviridae* (n = 5), *Parvoviridae* (n = 4), and *Caliciviridae* (n = 3), and other viral families (n = 8).

## The evolution and zoonotic potential of two SARS-related coronavirus

We next evaluated the evolutionary history and zoonotic potential of the two SARS-related coronaviruses detected in our samples: Bat SARS-like coronavirus CX1 and Bat SARS-like coronavirus LS1 (for simplicity referred to as SARS-like virus CX1 and SARS-like virus LS1, respectively, in the following text) (Fig. 6). Phylogenetic trees were estimated using the nucleotide sequences of key genes or domains: the RNA-dependent RNA polymerase (RdRp), N-terminal domain (NTD) and receptor-binding domain (RBD) of spike protein, and the nucleoprotein (N). This analysis revealed that in the NTD, RBD and N gene trees, SARS-like virus LS1 clustered with SARS-CoV forming an "S-1" clade, while SARS-like virus CX1 clustered with SARS-CoV-2 forming the "S-2" clade (Fig. 6a). Notably, while SARS-like virus LS1 remained in the S-1 clade in the phylogeny of the RdRp gene, SARS-like virus CX1 also fell into the S-1 clade. Hence, SARS-like virus CX1 appears to be a recombinant between the S-1 and S-2 lineages.

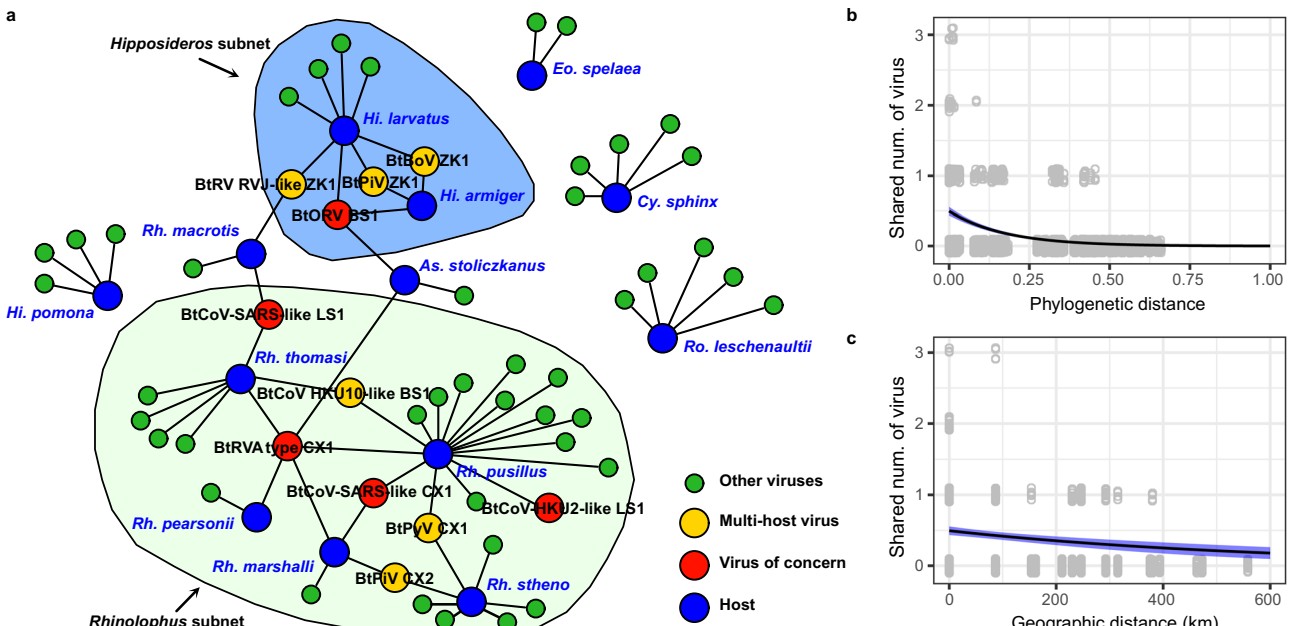

**Fig. 4 | The virus-sharing network of bats. a** The virus-sharing network reveals connectivity among viromes of different bat taxa. Viruses of concern and putative cross-species transmissions are shown in different colors. Two network modules (subnets) were detected with a network betweenness-based criterion and are visualized by colored areas. The relationship between the number of shared viruses with phylogenetic (**b**) or geographic distance (**c**) between pairs of host individuals. Phylogenetic distance is calculated as the sum of phylogenetic tree branch length between a pair of hosts, and the tree was estimated with nucleotide sequences of the COI gene employing a maximum likelihood method. The line and blue area mark the estimated partial effect and standard error of phylogenetic or geographic distance by Poisson regression. Blue shaded areas indicate 95% CI. Source data are provided as a Source Data file.

At the scale of the whole genome, SARS-like virus LS1 generally exhibited the highest genetic identity to human SARS-CoV viruses (93%). Indeed, in comparisons to previously identified SARS-related viruses (i.e., WIV16, Rs4231), SARS-like virus LS1 shared highest identity with human SARS-CoV viruses in ORF1b (nsp13 and nsp15) and the NTD, although it was relatively more distant in ORF1a and the RBD, as well as in the S2 domain of S gene (Supplementary Fig. 8). Specifically, it exhibited 98.13% similarity with SARS-CoV in the NTD, but only 88.61% identity in the RBD domain.

In marked contrast, SARS-like virus CX1 shared 92% genetic identity with SARS-CoV-2 at the whole genome scale, although with the occurrence of recombination. Indeed, we identified potential recombination at genomic positions 11,898–20,445 bp (3SEQ method, 99% CI: start position 11,876–11,937, end position 20,410–20,589), which encodes ORF1a (nsp7-nsp11) and ORF1b (nsp12-nsp14), with this region instead showing strong sequence similarity to SARS-CoV (92.3%). The remainder of its genome is very similar to SARS-CoV-2, particularly in the region encoding the NTD and RBD (95.15% and 93.70%, respectively), although no furin cleavage site was detected in the spike protein (Fig. 6b).

To evaluate the human-ACE2-receptor-binding potential of SARS-like virus CX1, we inferred the structure of its RBD using a homology-modeling approach and performed molecular dynamics simulations (Fig. 7). This revealed that there are only five amino acid substitutions in the RBD in comparison to the SARS-CoV-2 strain Wuhan-Hu-1 reference sequence, with three of these located at the interface of RBD-hACE2 complex (i.e., the receptor-binding motif) (Fig. 7a). Molecular dynamics simulations further revealed that the binding stability and energy of the RBD-hACE2 complex were very similar between SARS-like virus CX1 and SARS-CoV-2 Wuhan-Hu-1 (Fig. 7b and Supplementary Figs. 9 and 10).

In addition, we used in vitro assays to quantify the binding affinity of SARS-like virus CX1 to the human ACE2 receptor (Fig. 7). We first used an ELISA assay to show that the RBD of SARS-like virus CX1 can indeed bind to hACE2 receptor, despite of a lower affinity in comparison to SARS-CoV-2 (Fig. 7c). We then quantified the dissociation constant ($K_D$) of RBD-hACE2 complex using a biolayer interferometry (BLI) assay (Fig. 7d). The $K_D$ is 10.7 nM for SARS-like virus CX1, 0.26 nM for SARS-CoV-2 and 0.55 nM for SARS-CoV. In sum, these in silico and in vitro assays both indicate that SARS-like virus CX1 may be able to utilize human ACE2 receptor for cell entry.

## Discussion

We have characterized the mammal-associated virome of individual bats from China. This revealed an unexpectedly high frequency of virus co-infection, with 42% of the virus-positive individuals simultaneously infected by two or more viruses. The frequency of co-infection in individual bats has seldom been investigated, and only a few studies have explored the co-infection of specific viral species using consensus PCR methods (e.g., paramyxoviruses[19]). As such, this study provides evidence for virus co-infection using an unbiased omics approach. Co-infection is a prerequisite for virus recombination or reassortment[8], and the gut microbiome can facilitate the recombination of enteric viruses[20]. Hence, the high frequency of co-infection observed here suggests that recombination and reassortment are very likely to occur within individual bats, which in turn may facilitate the emergence of zoonotic viruses[9].

Our results also revealed frequent virus spillover among different bat species, identifying ten different viral species from different families that infect multiple host species. The ability of viruses to jump host species boundaries appears to be a near universal trait among viruses[21]. The frequent virus spillover among phylogenetically related or spatially co-located bats provides an opportunity for viromes of different bat species to exchange, further expanding genetic diversity of circulating viruses. Our results are of note because they show that the probability of virus spillover among pairs of host individuals is negatively associated with host phylogenetic and geographic distance, supporting the hypothesis that phylogenetically related or spatially closely located hosts share more viruses[22,23]. A specific example of this is the cross-species transmission of Bat SARS-like coronavirus CX1

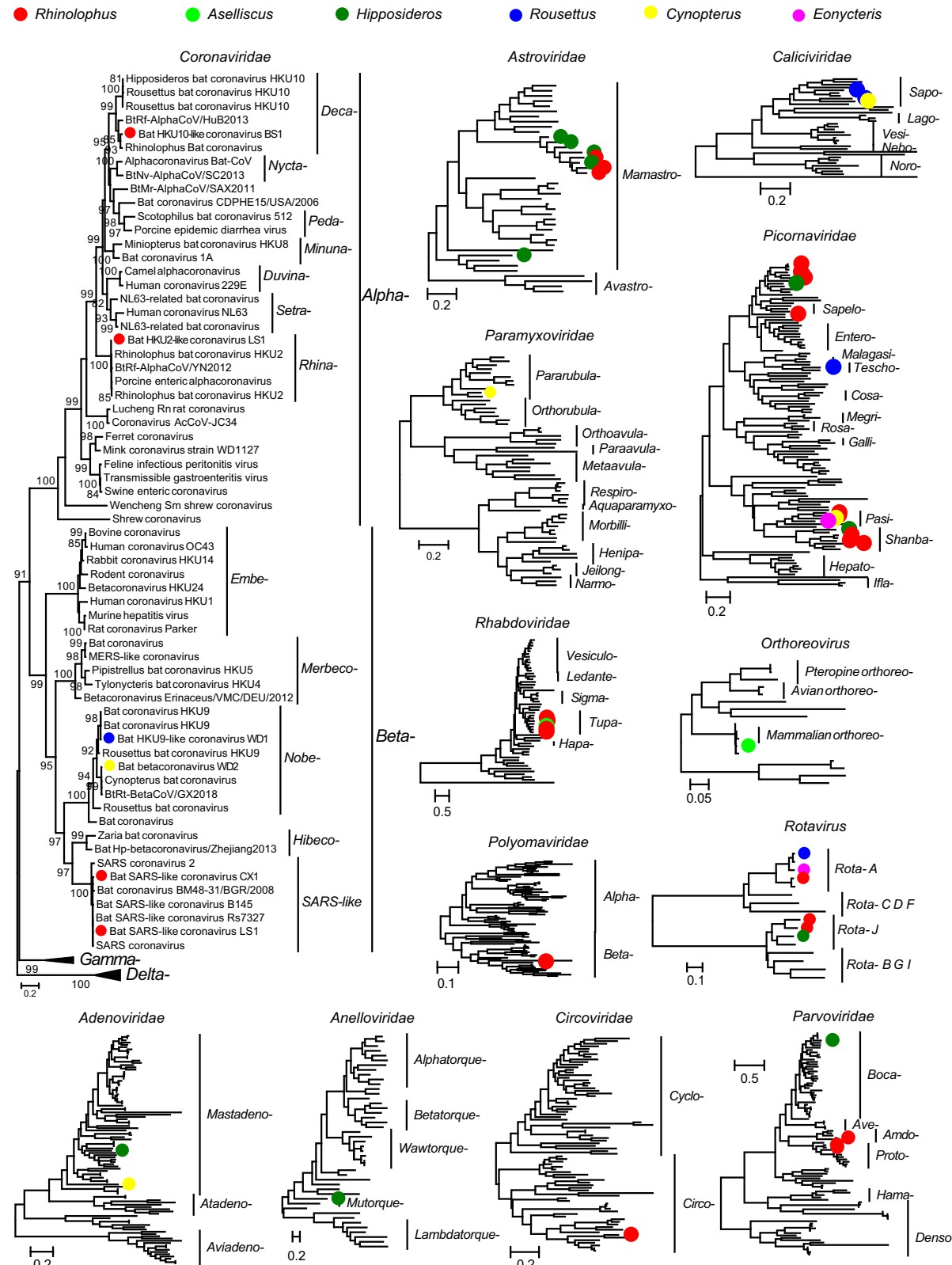

**Fig. 5 | Evolutionary relationships of 11 viral families detected in our study.** Phylogenetic trees were estimated using a maximum likelihood method based on conserved replicase protein (RNA viruses: RdRp, *Polyomaviridae*: LTAg, *Anelloviridae*: ORF1 protein, *Parvoviridae*: NS1, and other DNA viruses: DNA pol). Trees were midpoint rooted, and the branch length indicates number of nucleotide substitutions per site. Dots indicate viruses detected in our samples, and colors represent host genus.

**Table 1 | Identification of five "viruses of concern" and their prevalence among bats**

| Virus name | Genome identity to known human or livestock pathogens | Bat host species | Prevalence (positive/total individuals) | Genome size | |
|---|---|---|---|---|---|
| Bat SARS-like coronavirus LS1 | SARS-CoV Tor2 93.1% | *Rh. thomasi* | 2/14 | 29,756 bp | Complete genome |
| | | *Rh. macrotis* | 1/2 | | |
| Bat SARS-like coronavirus CX1 | SARS-CoV-2 Wuhan-Hu-1 92.3% | *Rh. pusillus* | 1/16 | 29,844 bp | Complete genome |
| | | *Rh. marshalli* | 1/7 | | |
| Bat HKU2-like coronavirus LS1 | SADS-CoV isolate 162140 93.6% | *Rh. thomasi* | 1/14 | 3087 bp | Partial genome |
| Bat orthoreovirus BS1 | Porcine reovirus SHR-A 69.2%~94.1% (Segments) 86.0% (Whole genome) | *As. stoliczkanus* | 3/35 | 23,245 bp | Complete genome |
| | | *Hi. armiger* | 3/11 | | |
| | | *Hi. larvatus* | 1/13 | | |
| | | *Rh. marcrotis* | 1/2 | | |
| Bat rotavirus A type CX1 | Human rotavirus A RVA/Human-wt/CHN/M2-102/2014/G3P[3] 86.8%~95.5% (Segments) 90.6% (Whole genome) | *As. stoliczkanus* | 9/35 | 18,189 bp | Complete genome |
| | | *Rh. thomasi* | 1/14 | | |
| | | *Rh. pusillus* | 1/16 | | |
| | | *Rh. marshalli* | 1/7 | | |
| | | *Rh. pearsonii* | 1/2 | | |

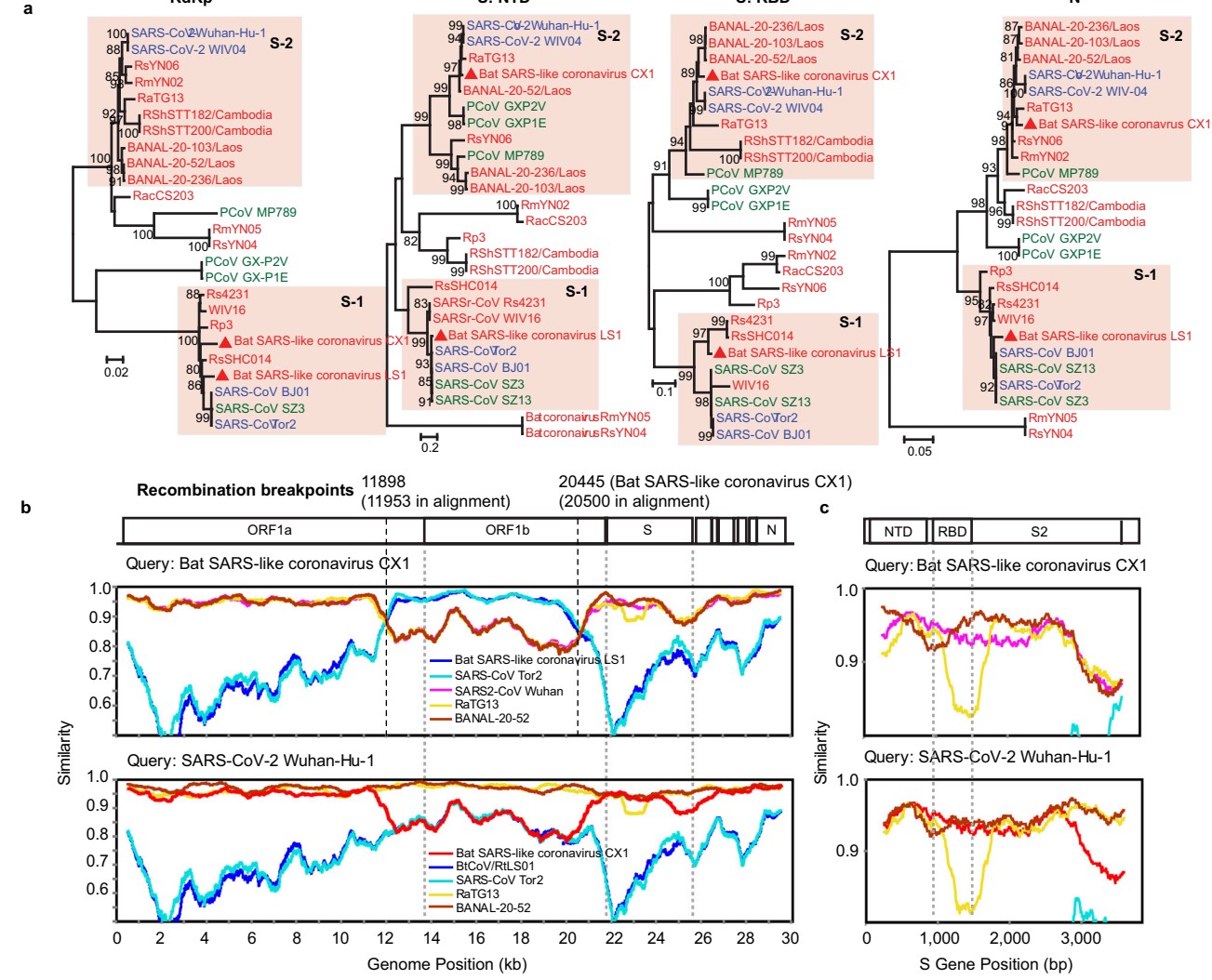

**Fig. 6 | Phylogenetic and structural analysis of a potentially zoonotic SARS-related coronavirus detected in our samples. a** Phylogenetic trees of four key functional genes of SARS-related coronaviruses. Colors of virus strain names indicate the host taxa where the viruses were detected. Red: bats, blue: human, green: others. Recombination analysis of SARS-related coronaviruses at the whole genome (**b**) and spike protein (**c**) scales. Source data are provided as a Source Data file.

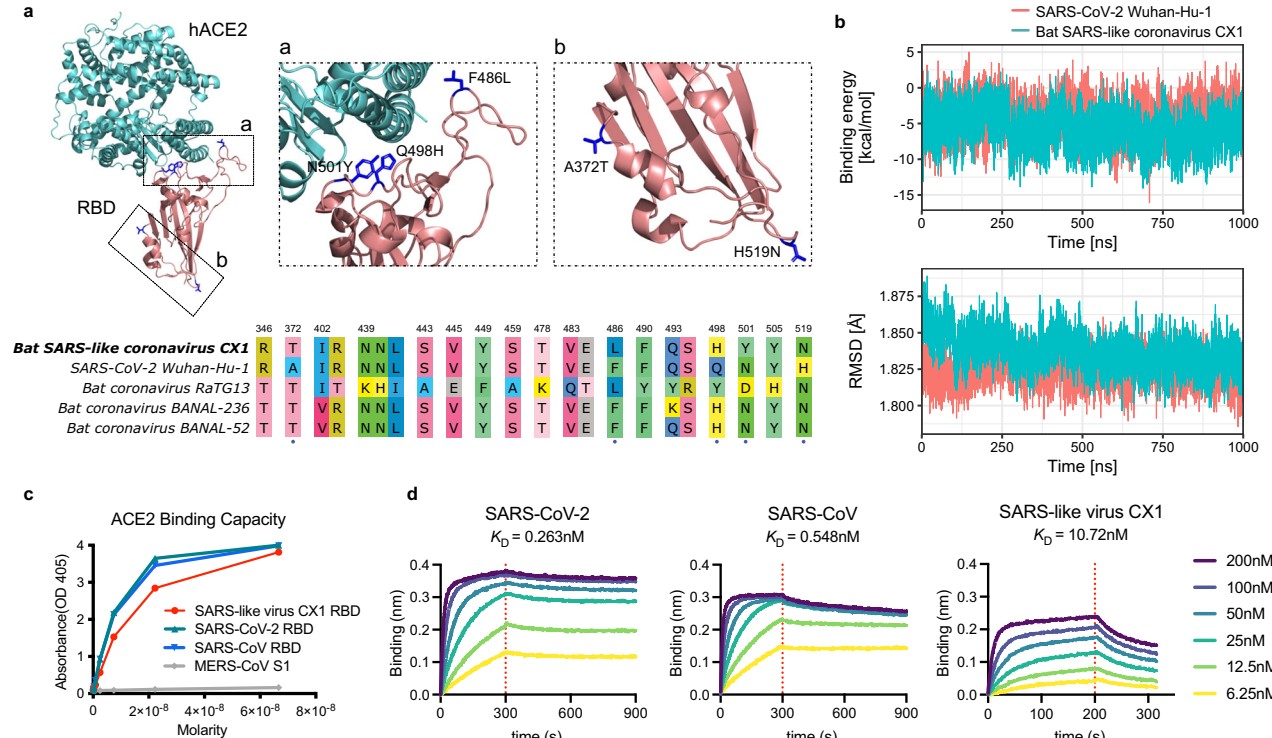

**Fig. 7 | In silico and in vitro assessment of the hACE2 receptor binding potential of a potentially zoonotic SARS-related coronavirus. a** Top, homology-modeling structure of the receptor-binding domain (RBD) of Bat SARS-like coronavirus CX1 in complex with human angiotensin-converting enzyme 2 (hACE2). Blue-colored residues on RBD indicate amino acid differences compared with SARS-CoV-2 Wuhan-Hu-1. Bottom, alignment of RBD sequences (residues T333 to G526 of spike protein) of Bat SARS-like coronavirus CX1, SARS-CoV-2 Wuhan-Hu-1 and two closely related bat coronavirus. Only polymorphic sites are shown. The five amino acid differences in the RBD of Bat SARS-like coronavirus CX1 compared to SARS-CoV-2 Wuhan-Hu-1 are marked with blue dots. **b** Molecular dynamics simulation results of binding energy (top) and binding stability (bottom) of Bat SARS-like coronavirus CX1 RBD-hACE2 complex. **c** The binding capability of SARS-CoV, SARS-CoV-2 and Bat SARS-like coronavirus CX1 RBD proteins to hACE2 protein was tested with various concentrations of the RBD proteins via ELISA. **d** The binding kinetics was determined by the biolayer interferometry (BLI) binding analysis. The purified hACE2 were coated on the sensor followed by the injection of various concentrations of SARS-CoV, SARS-CoV-2 and Bat SARS-like coronavirus CX1 RBD proteins. Source data are provided as a Source Data file.

between individuals of *Rh. pusillus* and *Rh. marshalli*, which were captured from the same cave at the same time. However, the cross-species transmission of viruses is limited by a number of host physiological and ecological barriers[22]. First, the virus might be able to infect both species in the absence of a high adaptive barrier. This likely involve binding to host receptors, cell entry, successful replication, and the evasion of host immunity. In this case, the two host species so are closely related that Bat SARS-like coronavirus CX1 is likely to readily overcome these barriers. Second, the two bat species co-habit the same cave, increasing transmission opportunities.

We identified two SARS-related coronaviruses in *Rhinolophus* bats (*Rh. marshalli*, *Rh. pusillus Rh. thomasi*, and *Rh. macrotis*) which we suggest are at particular risk for emergence. One of these—Bat SARS-like coronavirus CX1—is related to both SARS-CoV and SARS-CoV-2 and hence likely to have a history involving recombination. Notably, there are only five amino acid differences in the receptor-binding domain (RBD) of the spike protein of this virus compared to the Wuhan-Hu-1 reference genome of SARS-CoV-2[24], making it the closest relative to SARS-CoV-2 found in China in this particular genomic region. In contrast, the nsp7-nsp11 proteins of ORF1a and nsp12-nsp14 proteins of ORF1b were closely related to SARS-CoV, indicating that these genes were likely to be acquired from another SARS-related coronavirus. The remainder of the viral genome was closely related to SARS-CoV-2 and to several bat coronavirus previously found in Yunnan, including RaTG13[13], RmYN02[15], and RpYN06[14], all of which are close relatives of SARS-CoV-2. Together, these findings strongly suggest that virus spillover and co-infection in related bat species contribute to the

recombination of potentially pathogenic coronavirus and could possibly facilitate virus emergence in other species.

Functional analysis indicated that Bat SARS-like coronavirus CX1 likely has the ability to bind the human ACE2 receptor, albeit with lower affinity than SARS-CoV-2 Wuhan-Hu-1. Three of the five substitutions in the RBD—Q498H, N501Y and H519N—have been reported to increase affinity to human ACE2[25], and notably, the N501Y substitution is present in the Alpha, Beta, Gamma and Omicron variants of SARS-CoV-2. In addition, we found that the nsp7-nsp14 proteins (in which nsp12 is the replicase, i.e., RdRp) of Bat SARS-like coronavirus CX1 were closely related to those of SARS-CoV. A comparative study showed that SARS-CoV can replicate more rapidly than SARS-CoV-2 in vitro[26], while another suggested that nsp14 is likely associated with virulence[27]. These data tentatively suggest that the pathogenicity as well as virulence of Bat SARS-like coronavirus CX1 merits further consideration, and that this virus is potentially of high risk of emergence such that it should be monitored carefully.

We identified another four viruses of concern, likely to be pathogenic in humans or livestock. Bat SARS-like virus LS1 is closely related to SARS-CoV[28,29]. Bat HKU2-like coronavirus LS1 is closely related to SADS-CoV, which causes severe diarrhea and death in swine[30,31]. Bat rotavirus A type CX1 is related to human-infecting strains of Rotavirus A, which causes diarrhea[32,33], while Bat orthoreovirus BS1 is related to Mammalian orthoreovirus known to have a broad host range and cause diarrhea in swine[34,35]. Notably, although Rotavirus A is widespread in many bat genera as a single viral species, it remains questionable whether all genotypes or variants are able to infect a

board range of host species. Hence, zoonotic risk may vary among the genotypes of Rotavirus A depending on the genetic distance to known human-infecting strains[36]. Moreover, it is interesting that four of the five viruses of concern were found in more than one bat species in our samples, suggesting that these potentially zoonotic viruses may have a broader host range or have a higher rate of spillover than other viruses. However, whether frequent cross-species transmission among wild animals can translate into higher probability to emerge merits further mechanistic insights.

This study has a number of limitations. First, the sampling intensity was uneven among years, locations and bat species. Although our sampling design may be adequate for evaluating individual bat viromes, it could lead to biased estimates of total bat virome diversity at the population scale, as well as on the factors that contribute to cross-species virus transmission. Second, although we demonstrated that multiple viruses infected the same animal, recombination and reassortment require simultaneous infection of the same host cell, and analyses at this scale may require single-cell RNA sequencing. Third, in silico and in vitro assays of receptor binding do not necessarily reflect the pathogenicity of the viruses studied. Although our results showed that the RBD of Bat SARS-like coronavirus CX1 might be able to bind to hACE2, which implies that Bat SARS-like coronavirus CX1 may use hACE2 for cell entry, a more thorough assessment of the pathogenicity is still required.

In conclusion, using an individual virome approach we revealed a high frequency of virus co-infection and inter-species transmission among bats. The discovery of a diverse array of bat-associated viruses, including those potentially pathogenic to humans and livestock, emphasizes the need for continued vigilance in monitoring bat populations as potential sources of emerging infectious diseases. The identification of a novel recombinant SARS-like coronavirus that can utilize the human ACE2 receptor raises concerns about the potential for future zoonotic spillover events. Further research into the diversity and abundance of viruses within bats is necessary to better understand the risks associated with zoonotic transmission and to inform the development of strategies for disease prevention and control.

## Methods

### Ethics statement
This research, including the procedures and protocols of specimen collection and processing, was reviewed and approved by the Medical Ethics Committee of the Yunnan Institute of Endemic Diseases Control and Prevention (No. 20160002).

### Sample collection
A total of 149 rectum samples from bats were collected from six counties/cities in Yunnan province, China, during summer (July–August) and winter (November–December) between 2015 and 2019. The selection of bat sampling sites included the following factors: whether there were caves or other environments where bats were known to inhabit, the possibility of contact with humans, the presence of unexplained fever patients, and border areas. Specifically, bats were mainly collected from mountain caves (in Baoshan, Chuxiong, and Mengla), orchards (in Wanding), and crevices in the wild cliffs (in Lushui and Zhenkang). Bats collected from orchards have close contact with humans, those collected from mountain caves have occasional contact with a small number of people and animals, while those collected from crevices in the wild cliffs have minimal contact with humans. Bats were collected randomly by trapping with net traps and were primarily identified according to morphological criteria and confirmed by a barcode gene (COI) in the meta-transcriptomics analysis. The bats collected belonged to 15 species. The majority were from the genus *Rhinolophus* (n = 54) and comprised *Rhinolophus pusillus* (n = 16), *Rhinolophus thomasi* (n = 14), *Rhinolophus stheno* (n = 12), *Rhinolophus marshalli* (n = 7), *Rhinolophus pearsonii* (n = 2),

*Rhinolophus macrotis* (n = 2), and *Rhinolophus affinis* (n = 1). The genus *Hipposideros* (n = 26) animals comprised *Hipposideros larvatus* (13), *Hipposideros armiger* (11), and *Hipposideros pomona* (2). The genus *Rousettus* (n = 23) animals comprised *Rousettus leschenaultia* (n = 18) and *Rousettus amplexicaudatus* (n = 5). The *Aselliscus* (n = 35), *Cynopterus* (n = 9) and *Eonycteris* (n = 2) genera animals only contained *Aselliscus stoliczkanus* (n = 35), *Cynopterus sphinx* (n = 9), and *Eonycteris spelaea* (n = 2), respectively. All rectum samples were collected from each individual bat and then stored at −80 °C until use.

### RNA extraction, library preparation and sequencing
Each sample from individual bat were homogenized using grinding bowls and rods in MEM medium. The homogenized samples were then centrifuged at 8000 × g for 30 min at 4 °C to obtain supernatant. Total RNA extraction and purification were performed using the RNeasy Plus universal mini kit (QIAGEN) according to the manufacturer's instructions. RNA sequencing library construction and ribosomal RNA depletion were performed using the Zymo-Seq RiboFree™ Total RNA Library Kit (Zymo Research). Paired-end (150 bp) sequencing of the 149 dual-indexed libraries was performed on an Illumina NovaSeq platform.

### Viral genomes assembly and annotation
Raw paired-end sequence reads were first quality controlled, and rRNA reads were removed by mapping against the rRNA database downloaded from the SILVA website (https://www.arb-silva.de/) using Bowtie2. The clean reads were then de novo assembled into contigs using MEGAHIT (version 1.2.8)[37]. We performed a blastx search of contigs against the NCBI nr database using Diamond (version 0.9.25)[38] to roughly classify the sequences by kingdom. The e-value was set at 0.001 to achieve high sensitivity while reducing false positives. We identified viruses from assembled contigs based on hallmark genes (i.e., RdRp for RNA viruses *Polyomaviridae*: LTAg, *Anelloviridae*: ORF1 protein, *Parvoviridae*: NS1, and other DNA viruses: DNA pol), and we applied contig length filtering (contig length >1000 bp) as well as domain completeness (at least one conserved motif of RdRp should exist, checked manually by performing multi-sequence alignments) filtering as quality control. Those contigs classified as viruses were used for later analysis. Some viral contigs with unassembled overlaps were merged using SeqMan in the Lasergene software package (version 7.1)[39]. We searched for ORFs in each viral genome using the NCBI ORFfinder (https://ftp.ncbi.nlm.nih.gov/genomes/TOOLS/ORFfinder/), with the genetic code set to standard and with ATG as the only start codon. Then we performed a blastp search against the nr database and manually annotated the viral contigs according to the results.

### Viral species demarcation and phylogenetic analysis
Viral species were identified based using nucleotide sequences of whole genome or amino acid sequences of the conserved replicase proteins (RNA viruses: RdRp, *Polyomaviridae*: LTAg, *Anelloviridae*: ORF1 protein, *Parvoviridae*: NS1, and other DNA viruses: DNA pol). We applied a 90% cut-off of amino acid sequence and 80% nucleotides similarity to demarcate different virus species. If a viral species is at least 90% (hallmark protein) or 80% (whole genome) identical to existing viruses in GenBank, then it was characterized as a known virus species, otherwise it was denoted a novel virus species. Specifically, for species Rotavirus A, we determined its genotypes by comparing nucleotide similarity of all 11 segments with existing genotypes in the NCBI nucleotide database[40,41]. The viruses were then aligned using MAFFT (version 7.48)[42] and ambiguously aligned regions were removed using TrimAl[43]. Phylogenetic trees were then estimated by the maximum likelihood (ML) approach implemented in PhyML version 3.0[44], employing the LG model of amino acid substitution and the Subtree Pruning and Regrafting (SPR) branch-swapping algorithm. For SARS-related viruses, nucleotide sequences of RdRp, NTD, RBD and N

genes were used for phylogenetic analysis, employing the GTR substitution model.

## Quantification of virus abundance

We quantified the abundance of each virus in each library as the number of viral reads per million non-rRNA reads (i.e., RPM) by mapping clean non-rRNA reads of each library to the corresponding viral genomes. To reduce false positives, we masked low complexity regions (e.g., polyA tail) using bbmask tool (http://sourceforge.net/projects/bbmap/) and applied an abundance threshold of RPM > 1. The threshold has been validated in our previous publication[18] to achieve low false-positive rate. To reduce false positives due to index-hopping, we applied filter on read count of each virus[18]. If the total read count of a specific virus in a specific library of is less than 0.1% the highest read count for that virus within the same sequencing lane, then it is considered as a false positive due to index-hopping. These measures of quality-control could largely reduce the chance that sequences found in multiple bat samples were due to contamination during the sequencing process.

## PCR confirmation of virus genomes

To further confirm that sequences found in multiple bat samples were due to contamination during the sequencing process, the genome sequence of viruses of concern and a subset of viruses with low genome coverage (<30%) were obtained and confirmed by RT-PCR amplification and Sanger sequencing. Specifically for Bat SARS-like coronavirus CX1, the WTA product was performed using the Complete Whole Transcriptome Amplification Kit (WTA2)[45] (Sigma-Aldrich, St. Louis, MO), with the PCR reaction then performed using a set of self-designed primer pairs based on the obtained reads. To confirm the recombination breakpoints, long fragments of Bat SARS-like coronavirus CX1 were obtained using the SuperScript IV Reverse Transcriptase and Expand Long Template PCR System. The sequences of primers were presented in Supplementary Data 1.

## Recombination analysis of SARS-related viruses

Analyses of recombination among SARS-related viruses were performed using 3SEQ[46], and similarity plots were generated with Simplot 3.5.1[47]. A set of 77 *Sarbecovirus* which are closely related to SARS-like virus CX1 or LS1 were used for the initial recombination screen. Subsequently, the nucleotide sequences of the SARS-related viruses were analyzed with reference strains obtained from GenBank, comprising SARS-CoV Tor2, SARS-CoV-2 Wuhan-Hu-1, as well as most closely related bat SARS-related coronaviruses identified so far: Rs4231, WIV16, RaTG13, and BANAL-20-52.

## Homology modeling of the SARS-like virus CX1 RBD

We built homology models of the Bat SARS-like coronavirus CX1 RBD-hACE2 protein complex with MODELLER (version 10.3)[48], using the known structure of a SARS-CoV-2 RBD-hACE2 complex (PDB ID: 6M0J, resolution 2.45 Å)[49] as a template. The similarity between Bat SARS-like coronavirus CX1 RBD and the template was 97.4%. We removed all NAG and water molecules in the template, and kept the zinc and chloride atoms. We built 100 homology models and selected the top three models based on normalized DOPE score[50] for the later MD simulations.

## Molecular dynamics (MD) simulation

We used the CHARMM-GUI webservice[51] to prepare inputs for MD simulations. The three homology models described above and a SARS-CoV-2 RBD-hACE2 complex with known structure (PDB ID: 6M0J [RCSB PDB − 6M0J: Crystal structure of SARS-CoV-2 spike receptor-binding domain bound with ACE2]) were input to the CHARMM-GUI solution builder pipeline. The four systems were solvated in a water box of 13.5 nm × 9.2 nm × 8.3 nm, with KCl at the concentration of 0.15 M. We used CHARMM36m force field[52] for protein and ions, and TIP3P model[53] for water.

The models processed by CHARMM-GUI were then used as inputs to GROMACS (version 2022.3)[54,55] for MD simulations. The following steps were performed sequentially for each model: (1) energy minimization, (2) 1-ns-long equilibration in NPT ensemble, and (3) 1-ns-long equilibration in NVT ensemble. The temperature and pressure were set to 300 K and 1 atm, respectively. We then performed production simulations in NVT ensemble. Production simulation for the top homology model was 1000 ns long, and we performed another two 500-ns-long simulations for the remaining two homology models as replicates. Similarly, we performed one 1000-ns-long production simulation for the SARS-CoV-2 RBD-hACE2 complex (PDB ID: 6M0J) and two 500-ns-long replicates.

We performed two sets of analyses on the data retrieved from MD. First, we evaluated the stability of RBD-hACE2 binding by measuring deviation of the protein backbones (measured as RMSD) in the duration of simulations, using PLUMED (version 2.7.4)[56]. The backbone RMSD were calculated with respect to energy-minimized structure of each model. We also calculated RMSD separately for RBD, hACE2 and the RBD-hACE2 interface (residues within 0.8 nm to the other subunit in the 6M0J model). Second, we estimated and compared the binding energy of RBD-hACE2 complex using FoldX (version 4)[57]. We visualized the structure of RBD-hACE2 complex using PyMOL (version 2.4.2).

## In vitro synthesis and purification of SARS-like RBD

We synthesized and cloned the coding sequences of Bat SARS-like coronavirus CX1, SARS-CoV-2, and SARS-CoV RBD with a carboxy-terminal 8×His tag, and hACE2 with an Fc domain, into the pcDNA3.1 plasmid. After transfecting the plasmids into HEK293T cells (ATCC, catalog number: CRL-3216) with polyethylenimine (PEI) and culturing for 5 days, we purified the proteins using affinity chromatography with Ni-NTA or protein-A resin and assessed their purity with SDS-PAGE.

## ELISA assays of hACE2 binding potential

To perform these assays 96-well plates were coated overnight at 4 °C with 2 μg/ml of Bat SARS-like coronavirus CX1, SARS-CoV-2, and SARS-CoV RBD. After blocking, serial 3-fold diluted hACE2 was incubated for 1 h at 37 °C, starting at 10 μg/ml. Subsequently, the plates were washed with PBST (PBS containing 0.05% Tween-20). To detect the binding of hACE2 Horseradish peroxidase (HRP)-conjugated goat anti-human IgG antibody (polyclonal) diluted 1:2000 (Jackson ImmunoResearch, catalog number #109-035-098) was added and incubated for 1 h at 37 °C. The plates were developed with Super Aquablue ELISA substrate (eBiosciences), and the absorbance was measured at 405 nm using a microplate spectrophotometer (MolecularDevices).

## Biolayer interferometry

To determine the binding affinity, protein-A sensors were activated with a PBS buffer containing 0.02% Tween-20 and 0.1% BSA, followed by immobilization of 50 μg/ml hACE2-Fc protein onto the sensors. The biosensors were then exposed to varying concentrations of purified Bat SARS-like coronavirus CX1 RBD in the same buffer. The binding affinity, association rate, and dissociation rate were calculated using ForteBio software, and the data were plotted using Prism 9.

## Statistical analysis of cross-species transmission of viruses

All statistical analyses were performed in R (version 4.2.0). To reveal possible cross-species virus transmission events, we visualized the virus-sharing pattern among different bat species using a bipartite network. In this network, a node is either a host or a virus species, and an edge linking a host node and a virus node indicates the presence of that virus in that host. We performed edge betweenness clustering on

such network to find network modules, which are subset of nodes such that connections between these nodes are denser than outside of the subset, using the igraph package in R. A biological interpretation of a network module is that host species within the same module shared more viruses than outside that module.

We performed two sets of statistical tests to further quantitatively evaluate cross-species transmission of viruses among bats. First, we assessed the strength of the correlation of virome composition with both host phylogeny and geographic location using partial Mantel tests implemented in the ecodist package. Differences between virome compositions were represented by Bray–Curtis distance, and phylogenetic distance between hosts was measured as the sum of branch length of pairs of hosts in the COI gene tree. The intervals between sampling dates were included in the partial Mantel tests to exclude the confounding effect of time/seasons. We then used Poisson regression to estimate the effects of (1) phylogenetic distance between hosts, and (2) geographic distance between sample locations on the number of shared virus species between pairs of hosts, including the time intervals between sampling dates to control for its confounding effect.

### Reporting summary

Further information on research design is available in the Nature Portfolio Reporting Summary linked to this article.

## Data availability

The meta-transcriptomic sequencing reads generated in this study have been deposited in the SRA database under accession code PRJNA929070. The viral genome sequences generated in this study have been deposited in the NCBI GenBank database under accession code (OP963575-OP963684, OQ709180-OQ709197 and OQ934005-OQ934007) and China National GeneBank DataBase, i.e., CNGBdb (project accession: CNP0003916). The sequence data deposited in NCBI and CNGBdb are identical. The sample metadata and other materials generated in this study are provided in the GitHub repository (https://github.com/Augustpan/Individual-Bat-Virome, https://doi.org/10.5281/zenodo.7941061). Source data are provided with this paper.

## Code availability

Code and scripts are provided in a GitHub repository (https://github.com/Augustpan/Individual-Bat-Virome, https://doi.org/10.5281/zenodo.7941061).

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

## Acknowledgements

We thank the Alibaba Cloud Computing Co. Ltd. for providing the computational resources for rapid data processing, and the local Centers for Disease Control and Prevention in six trapping sites for their assistance in specimen collection. M.S. was supported by Shenzhen Science and Technology Program (KQTD20200820145822023), Guangdong Province "Pearl River Talent Plan" Innovation and Entrepreneurship Team Project (2019ZT08Y464), and Health and Medical Research Fund (COVID190206). Y.F. was supported by the National Natural Science Foundation of China (31560049), Yunnan Reserve Talents for Academic and Technical Leaders of Middle-Aged and Young People (2019HB052). E.C.H. was supported by a National Health and Medical Research Council (Australia) Investigator grant (GNT2017197) and by InnoHK funding from the Innovation and Technology Commission of Hong Kong. Y.Q.C. was supported by Shenzhen Science and Technology Program (RCJC20210706092009004). G.L. was supported by United States National Institutes of Health U01 AI151810. S.C. was supported by the National Natural Science Foundation of China (32171192).

## Author contributions

Conceptualization: E.C.H., Y.F. and M.S.; Methodology: J.W., Y.-F.P., W.-H.Y., K.L., C.-M.L., W.-C.W., B.L., S.C., Y.-L.S., D.G., G.L., J.L., Y.F., E.C.H. and M.S.; Investigation: J.W., Y.-F.P., L.-F.Y., W.-H.Y., K.L., C.-M.L., J.W., G.-P.K., W.-C.W., Q.-Y.G., G.-Y.X., H.-L.L., S.C., Y.-Q.C., E.C.H., Y.F. and M.S.; Writing—original draft: Y.-F.P.; Writing—review and editing: J.W., Y.-F.P., L.-F.Y., W.-H.Y., C.-M.L., J.W., G.-P.K., W.-C.W., Q.-Y.G., G.-Y.X., H.-L.L., Y.-Q.C., Y.-L.S., D.G., Z.-H.G., G.L., J.L., E.C.H. and M.S. Funding acquisition: J.L., Y.F., and M.S.; Resources (sampling): L.-F.Y., W.-H.Y., Z.-H.G., G.L. and Y.F.; Resources (computational): J.L., E.C.H. and M.S.; Supervision: Z.-H.G., G.L., Y.-Q.C., E.C.H., Y.F. and M.S.

## Competing interests

The authors declare no competing interests.

## Additional information

¹State Key Laboratory for Biocontrol, School of Medicine, Shenzhen Campus of Sun Yat-sen University, Sun Yat-sen University, Shenzhen, China. ²Shenzhen
Key Laboratory for Systems Medicine in Inflammatory Diseases, Shenzhen Campus of Sun Yat-sen University, Sun Yat-sen University, Shenzhen, China.
³Ministry of Education Key Laboratory of Biodiversity Science and Ecological Engineering, School of Life Sciences, Fudan University, Shanghai, China.
⁴Department of Viral and Rickettsial Disease Control, Yunnan Provincial Key Laboratory for Zoonosis Control and Prevention, Yunnan Institute of Endemic
Disease Control and Prevention, Dali, Yunnan, China. ⁵School of Public Health (Shenzhen), Shenzhen Campus of Sun Yat-sen University, Sun Yat-sen
University, Shenzhen, China. ⁶Yunnan Key Laboratory of Plant Reproductive Adaptation and Evolutionary Ecology and Centre for Invasion Biology, School of
Ecology and Environmental Science, Yunnan University, Kunming, Yunnan, China. ⁷Molecular Imaging Center, Central Laboratory, The Fifth Affiliated
Hospital, Sun Yat-sen University, Zhuhai 519000 Guangdong, China. ⁸Guangzhou National Laboratory, Guangzhou International Bio-Island, Guangzhou,
Guangdong Province, China. ⁹State Key Laboratory of Infectious Disease Prevention and Control, National Institute for Viral Disease Control and Prevention,
Chinese Center for Disease Control and Prevention, Beijing, China. ¹⁰Department of Infectious Diseases and Public Health, Jockey Club College of Veterinary
Medicine and Life Sciences, City University of Hong Kong, Hong Kong, China. ¹¹Sydney Institute for Infectious Diseases, School of Medical Sciences, The
University of Sydney, Sydney, NSW 2006, Australia. ¹²These authors contributed equally: Jing Wang, Yuan-fei Pan, Li-fen Yang.
✉e-mail: chenyaoqing@mail.sysu.edu.cn; edward.holmes@sydney.edu.au; ynfy428@163.com; shim23@mail.sysu.edu.cn

