## [Peer Review File · Nature Communications]

Individual bat viromes reveal the co-infection, spillover and emergence risk of potential zoonotic virusesREVIEWER COMMENTS

Reviewer #1 (Remarks to the Author):

This manuscript describes the detection of mammalian viruses in individual bats sampled in Yunnan province, China. The authors find a high frequency of virus co-infection in individual bats and the presence of multiple viruses shared among several bat species, suggesting possible spillover and the possibility of viral recombination/reassortment. They also examine the individual viral species found in these bats and note viruses with high similarity to known human and livestock viruses, since these species may have the highest likelihood of spilling over into human populations.

The manuscript is well written and addresses a very important topic — the need to better understand viral diversity and transmission patterns in bat populations, including patterns of co-infection and possible host spillover — and provides important data collected from individual bats. However, I found it difficult to evaluate some of the conclusions made by the authors from the data presented and my main concern is that the number of sampled bats may not be sufficient to make claims about virus prevalence in particular bat species. I would like to see this manuscript published since I agree with the authors that details on individual bat viromes are lacking, but I feel that a significant revision would be required to ensure the findings are indeed valid and reproducible.

(1) Could the authors provide any information on how representative the 149 bat individuals are of the locations/years in which they are collected? For example, I see from the supplemental data that there were very few bats collected in 2019, and that all these bats were from the same sampling site. Do the authors know much about the expected bat diversity at each site, and how much bat geography is expected to change from year to year? Furthermore, while I was able to find some details about the 149 bat individuals in the supplementary material, a clear summary of this data in the main text (e.g., how many bats per year per site) would aid interpretation of later analyses (e.g., adding a time element to Figure 1).

(2) Could the authors provide more details on the sequencing data generated, including if controls for contamination were used and how many reads were generated per sample? How confident were the authors in viral species assignments? How good was coverage of the various viruses detected? Were only fragments of viruses detected, or were the authors able to generate full genome assemblies, and what was the read depth of these assemblies? These details would be very helpful to understanding the abundance analyses.

(3) Could the authors provide some details on the key differences between *Rhinolophus* and *Aselliscus* bats? This would help me understand if the finding of more virus species per individual in *Rhinolophus* bats (line 125) makes sense or is potentially due to sampling error.

(4) How did the authors control for index hopping (line 129)? Were other sequencing controls or quality checks used to ensure high quality results?

(5) The authors claim that, based on the 12 virus species that were found to be shared across species, certain viruses have broader host range (line 131). How do the authors know that this is not a sampling artefact? Is it possible that the sequencing of more bats from one particular species gave more opportunities to find a certain virus in that species? In Figure 3, the authors show the prevalence of each viral family among different bat species, and there seems to be a correlation between the number of positive bat individuals for a particular virus and the prevalence of the viral species in that host. This suggests to me that this could be a sampling artefact, as sequencing of more individuals of a species led to more virus findings. I think it is fair to present the virus findings (e.g., as in Table S5), but claims about host range and relative prevalence (e.g., line 152) should be toned down.

(6) In Figure 4B-C, did the authors perform the analysis at the individual level or did they aggregate results across species and then look for geographic and phylogenetic associations? I ask because it would be important to control for sampling of multiple bat species from the same place, for example, as this could bias the geographic distance correlation.

Minor comments for the authors to consider:

Line 95: Could the authors define “mammal-associated” virome? The authors define the term later (line 111) but it would be helpful to have the definition sooner.

Figure 1B: Are all bat species included in this tree? As a non-bat expert this would be helpful to know.

Line 118: I find “virus load” to be a confusing term to use throughout this section, as this is usually something that is determined experimentally. Without details on the read depth and distribution across the genome (see above comment), it is difficult to know if the number of mapped reads truly correlates to viral load, correct?

Figure 5: Could the authors highlight the viruses of concern more clearly in this figure?

Line 170: Are the 28 viral species newly discovered viruses, or previously discovered viruses now found in mammals for the first time?

Figure 6B: Why is this figure split into the top and bottom figure panels? Would it not make sense to compare all viruses on the same plot? If the figures are split into two, consistent colors for each virus would be helpful.

Reviewer #2 (Remarks to the Author):

The manuscript by Jing Wang et al. presented an interesting approach to look at virome at individual bat level. The result is very interesting showing a large diversity of viruses carried by different bat species and a frequent virus sharing among bat species. Interestingly, the authors discovered a recombinant SARS-CoV- like virus with evidence of history of recombination with ability to utilize human ACE2 receptor for cell entry. The manuscript is well thought and written. There are some small points to be clarified.

- What does BtSY stand for? BtSY seems to be associated to different viruses. Using BtSY1 and BtSY2 may not be appropriate to represent SARS-like virus. For example, line 136 has Bat RVJ-like rotavirus BtSY1.
- Lines 154, though we can find the information in the figure 4, it would nice to give some context of the detection of the two SARS-like virus in bats. What were the bat species? Were the virus detected in multiple bats?
- Rotavirus seems to be predominantly detected in different bat genus. Do authors have any explanation? It would be interesting to develop a bit more in discussion. Rotavirus A is the most common species to cause infection in human. How close the virus detected in bats to the ones that cause disease in human?
- Line 264, Sample collection. Were the capture done in open air or at the entrance of cave? Do the locations present any interfaces with human or other animals? How the locations were selected?
- Although the in silico tools used to evaluate human ACE2 binding potential of BtSY2 gave strong evidence showing BtSY2 may utilize human ACE2 receptor for cell entry, I wonder why authors did not perform in vitro experiment which is pretty common in the first place.

Reviewer #3 (Remarks to the Author):

The work presented by Wang et al. described the mammalian virome of individual bat rectal swabs collected over a 4-year period in Yunnan, China. The findings of the study are not very innovative compared to other studies, but sarbecoviruses described here are of interest, especially regarding the origin of SARS-CoV-1 and SARS-CoV-2 viruses.

Major comments:

1. The mention “likely to be pathogenic for humans and livestock” (regarding coronaviruses) present in the abstract and through the whole manuscript is a bit speculative and need to be mitigated. The conservation of contact RBD/ACE2 residues and MD simulations give some evidence of a possible human infection but do not inform on the putative pathogenicity of the virus for humans or livestock.
2. Co-infection is a pre-requisite for virus recombination, and I agree with the authors that the study of individual virome is important to study co-infection. However, recombination can occur only when a given cell is concomitantly infected by at least two viruses; and the description of viral communities of a bat rectal swab does not inform on which cell is infected. So I strongly suggest the authors to add a paragraph to discuss this issue and to change the titer because their study does not provide rational

evidence of co-infection.

3. Regarding the cross-species transmission of viruses among bats, an important information has to be added in the manuscript: when the authors claim that the same virus species is present in different bat species, did they perform a distance matrix (or another type of analysis) to compare the genetic distance/identity between the different strains of a given virus species identified in the different bats? For example, is it the same BtSY2 coronavirus (and how close are these strains) that is shared between *R. pusillus* and *R. marshalli*? Such analysis will give additional evidence of the cross-species transmission of the same virus to different bat species. And if it the case, how do the authors explain this cross-species transmission: co-living in the same caves? Another explanation?

4. Following this point of cross-species transmission of viruses among bats, I am not sure that the network analysis presented in Figure 4 is suitable and sufficient to claim that “more closely phylogenetically related or closely geographically located bat individuals had more similar virome compositions and had more virus species in common” since the analysis was restricted to mammalian-infecting viruses. Could the authors perform a more comprehensive description of the individual virome that include viruses associated to prokaryotes, plants, etc.? It will help to distinguish the core virome (shared by different bat species/genera) and the virome which is specific to a given bat species or genus.

5. The results regarding the BtSY2 coronavirus are interesting, but a detailed recombination analysis is needed (with for example GARD or 3SEQ). SimPlot is not a tool to determine recombinant breakpoints. How do the authors determine the coordinates of breakpoint mentioned line 193 page 7? In addition, the MD simulation and the analysis of mismatches in the ACE2/RBD contact residues are not sufficient to claim that “these data tentatively suggest that BtSY2 may be able to replicate rapidly with similar virulence as SARS-CoV”. It is too speculative.

Other comments:

1. A table (or the update of Table S5) should be added to present the horizontal genome coverage achieved for the viruses of concern, with the description of the closest viral sequence and the % identity in the different genes. In fact, we do not know in the manuscript if the authors obtained the full genomes for these viruses of concern.

2. Please add a conclusion of the study.

3. Do the authors have studied the effect of season or gestation stage of bats o sample location on the composition of viromes? These extrinsic factors may play an important role in the difference observed in virus abundance and diversity.

Minor comments:

1. Introduction line 86 page 4: “A number of highly pathogenic viruses have been detected there”: give some example, because bat-borne relatives of SARS-CoV-1 and SARS-CoV-2 are not “highly pathogenic”

2. Results line 120 page 5: “We consider those with relatively high viral load (reads per million total reads > 1) as true positives”. How do you defined this threshold?

3. Line 148 page 6: typo “what” instead of “that”

4. Line 158 page 6: NTD and RBD are not genes but functional domains of the spike

5. Line 167 page 6 “The nucleotide identity between their RdRp was less than 80%, so we demarcated them as three different types”. Please use the ICTV demarcation criteria.

6. Line 241 page 8 “and even has slightly higher affinity than SARS-CoV-2 Wuhan-Hu-1”: how do the authors claim that?
7. Figure 6B: for a better comprehension, I would rather present SARS-CoV-2 Wuhan as a reference for Simplot analysis at the whole genome level. And please use the same strains for Simplot analyses at the whole genome and at the spike levels: reader may not know all the strains used here.
8. Figure 6B, legend: “Recombination analysis”. NO, SimPlot is not a recombination analysis tool.
9. Table S3 is not referred in the text

SUMMARY OF THE MAJOR REVISIONS TO THE MANUSCRIPT

1. We performed two *in vitro* assays (ELISA assay and BLI assay) to quantify the receptor-binding affinity of the Bat SARS-like coronavirus CX1 (renamed from SARS-like coronavirus BtSY2).
2. We added 14 bat-associated viral species that were mistakenly omitted from the previous analysis. None of the 14 viral species were found in multiple host species (i.e., cross-species transmission), and none are closely related to known pathogens (i.e., ‘viruses of concern’).
3. We renamed all viral species based on consistent criteria. For example, Bat SARS-like coronavirus BtSY2 is renamed to Bat SARS-like coronavirus CX1 (or SARS-like virus CX1 in a shortened form), as it was identified in Chuxiong country, Yunnan.
4. We provided a total virome analysis (including those viruses associated with bat microbiome or diet) data in the supplementary materials as requested by Reviewer 3.

Other revisions are also made, please see the point-by-point response below:

Note: We present all revision to the manuscript in using “Track-Change mode” in MS Word. This mode records line number of deleted content, so the line numbers may look discontinuous, but the actual contents are continuous indeed.

REVIEWER COMMENTS

Reviewer #1 (Remarks to the Author):

This manuscript describes the detection of mammalian viruses in individual bats sampled in Yunnan province, China. The authors find a high frequency of virus co-infection in individual bats and the presence of multiple viruses shared among several bat species, suggesting possible spillover and the possibility of viral recombination/reassortment. They also examine the individual viral species found in these bats and note viruses with high similarity to known human and livestock viruses, since these species may have the highest likelihood of spilling over into human populations.

The manuscript is well written and addresses a very important topic — the need to better understand viral diversity and transmission patterns in bat populations, including patterns of co-infection and possible host spillover — and provides important data collected from individual bats. However, I found it difficult to evaluate some of the conclusions made by the authors from the data presented and my main concern is that the number of sampled bats may not be sufficient to make claims about virus prevalence in particular bat species. I would like to see this manuscript published since I agree with the authors that details on individual bat viromes are lacking, but I feel that a significant revision would be required to ensure the findings are indeed valid and reproducible.

(1) Could the authors provide any information on how representative the 149 bat individuals are of the locations/years in which they are collected? For example, I see from the supplemental data that there were very few bats collected in 2019, and that all these bats were from the same

sampling site. Do the authors know much about the expected bat diversity at each site, and how much bat geography is expected to change from year to year? Furthermore, while I was able to find some details about the 149 bat individuals in the supplementary material, a clear summary of this data in the main text (e.g., how many bats per year per site) would aid interpretation of later analyses (e.g., adding a time element to Figure 1).

Response: We added a bar plot to Figure 1A to show the number of samples per year, per site. The sample size we collected was limited due to research budget constraints, and therefore the sample did not cover all bat species in Yunnan province. However, according to the bat species diversity data in Yunnan (China Species Library, <https://species.sciencereading.cn/>), we believe that we have collected the dominant bat species from each county (we randomly collected these bats so that the majority of captured individuals would be dominant species), such that it is representative. Details of the sampling methods used are described between lines 701-710 (Methods section). As for the bat geography issue, in most cases, the distribution range of bats is stable between years. Although some bat populations may undergo seasonal migration or be affected by human activities, range of these migrations are usually limited. We agree that the uneven sampling intensity may produce biased results in our analyses of diversity and patterns of cross-species transmission. We have therefore added a "Limitations" section (lines 664-674) to the Discussion which covers these points.

(2) Could the authors provide more details on the sequencing data generated, including if controls for contamination were used and how many reads were generated per sample? How confident were the authors in viral species assignments? How good was coverage of the various viruses detected? Were only fragments of viruses detected, or were the authors able to generate full genome assemblies, and what was the read depth of these assemblies? These details would be very helpful to understanding the abundance analyses.

Response: We added relevant details on the bioinformatics pipeline to the first paragraph of the Results (between lines 154-160) and Methods (between lines 736-743) sections, especially how viral species are identified and quality-controlled. Briefly, we identify viruses from meta-transcriptomics assembled contigs based on hallmark genes (e.g., the RdRp for RNA viruses), and we applied contig length filtering as well as domain completeness filtering as quality controls. Finally, we cluster viral contigs at 90% amino-acid or 80% nucleotide identity to demarcate virus species. If a viral species is at least 90% (hallmark protein) or 80% (whole genome) identical to existing viruses in GenBank, then it is considered to be a known viral species, otherwise it is characterised as a novel virus species.

We have also added a new supplemental Table S3 to describe the completeness, read depth and coverage of all viral genomes. For the five viruses of concern, we added a new Table 1 to the main text that shows the genome completeness and their relatedness to known viruses.

(3) Could the authors provide some details on the key differences between *Rhinolophus* and *Aselliscus* bats? This would help me understand if the finding of more virus species per individual in *Rhinolophus* bats (line 125) makes sense or is potentially due to sampling error.

Response: The physiological (e.g., immune system) or ecological (habitat type, community size, etc.) differences among bat species/genera may potentially lead to differences in the number of viruses per host. Although this is certainly a very interesting area, it possible that sampling

biases when comparing the number of viral species per individual among species/genera will have impacted our analysis. We have therefore removed the Poisson regression (the right panel of original Figure 3B), leaving only the bar plot (new Figure 3b). We also added a “Limitations” section to the Discussion (lines 664-674) that explores the potential impact of sampling design on these results. While we believe that this individual-resolution data is valuable, the sampling design of our study may confound among-genera comparisons.

(4) How did the authors control for index hopping (line 129)? Were other sequencing controls or quality checks used to ensure high quality results?

Response: To ensure high-quality results, we implemented multiple measures throughout our viral genome assembly and species identification processes. Specifically, we controlled for contig length and RdRp domain completeness, and applied index hopping controls and masking of low-complexity regions during viral abundance quantification. Additionally, we utilized RT-PCR to confirm the presence of a subset of viruses with relatively low read coverage. These quality control measures have been validated and successfully utilized in our prior publications (e.g., Shi et al., 2022). Detailed descriptions of the specific methods used can be found in the Materials and Methods section, lines 769-775.

Shi, M., Zhao, S., Yu, B., Wu, W. C., Hu, Y., Tian, J. H., ... & Zhang, Y. Z. (2022). Total infectome characterization of respiratory infections in pre-COVID-19 Wuhan, China. *PLoS Pathogens*, 18(2), e1010259.

(5) The authors claim that, based on the 12 virus species that were found to be shared across species, certain viruses have broader host range (line 131). How do the authors know that this is not a sampling artefact? Is it possible that the sequencing of more bats from one particular species gave more opportunities to find a certain virus in that species? In Figure 3, the authors show the prevalence of each viral family among different bat species, and there seems to be a correlation between the number of positive bat individuals for a particular virus and the prevalence of the viral species in that host. This suggests to me that this could be a sampling artefact, as sequencing of more individuals of a species led to more virus findings. I think it is fair to present the virus findings (e.g., as in Table S5), but claims about host range and relative prevalence (e.g., line 152) should be toned down.

Response: We agree that the results on the observed host range (number of host species) of viruses may be confounded by uneven and inadequate sampling. We therefore toned down the claims about host range and relative prevalence (for example, by removing any reference to “the broadest host”). While acknowledging the possibility of sampling bias, it is worth noting that previous reports also suggest that certain viral species, such as Rotavirus A and Mammalian orthoreovirus, are more likely to have broad host ranges (please see Discussion, lines 652-663). Our findings align with this evidence, indicating that the observed patterns may not solely be a result of our sampling approach.

(6) In Figure 4B-C, did the authors perform the analysis at the individual level or did they aggregate results across species and then look for geographic and phylogenetic associations? I ask because it would be important to control for sampling of multiple bat species from the same place, for example, as this could bias the geographic distance correlation.

Response: We performed the Poisson regression and partial Mantel tests at the level of individual animals (i.e., we calculated phylogenetic distance, and geographic distance for each pair of bat individuals, and did regression based on these data). Performing these analyses among pairs of individuals is less likely to be confounded by the uneven number of samples. In addition, we controlled for the confounding effect of date of sampling (lines 904-914).

Minor comments for the authors to consider:

Line 95: Could the authors define “mammal-associated” virome? The authors define the term later (line 111) but it would be helpful to have the definition sooner.

Response: The “mammal-associated” virome refers to viruses that are likely to infect bats based on phylogenetic relatedness to known viruses of mammalian hosts, in contrast to those viruses associated with bat microbiome or diet (rather than the bats themselves). We have revised the text accordingly (lines 130-141).

Figure 1B: Are all bat species included in this tree? As a non-bat expert this would be helpful to know.

Response: This tree does not include all bat species. We manually picked a subset of reference sequences of a typical barcoding gene (Cytochrome c oxidase subunit I, COI, a mitochondrial gene) from NCBI to estimate the phylogeny. We also provided a full phylogenetic tree of bats in Fig.S1.

Line 118: I find “virus load” to be a confusing term to use throughout this section, as this is usually something that is determined experimentally. Without details on the read depth and distribution across the genome (see above comment), it is difficult to know if the number of mapped reads truly correlates to viral load, correct?

Response: Thank you for pointing out this. We have misused the concept of “viral load” and we did not measure viral load directly in this study. We have therefore changed “viral load” to “viral abundance” (measured as the number of viral reads per million clean reads, RPM) throughout the text.

Figure 5: Could the authors highlight the viruses of concern more clearly in this figure?

Response: Revised as requested.

Line 170: Are the 28 viral species newly discovered viruses, or previously discovered viruses now found in mammals for the first time?

Response: Novel viruses mean viruses that have not been previously discovered in any location. We have reworded the sentence to remove ambiguity (lines 475), and added a definition of novel virus in the Results (lines 476): that is, viruses with < 90% amino acid identity (hallmark genes, such as the RdRp) or 80% nucleotide identity (whole genome) to any existing virus genomes/proteins in the NCBI nucleotide or protein database.

Figure 6B: Why is this figure split into the top and bottom figure panels? Would it not make sense to compare all viruses on the same plot? If the figures are split into two, consistent colors for each virus would be helpful.

Response: These plots were originally used to reflect the potential recombination history (indicated by the intersection of similarity curves) of the two SARS-like coronaviruses separately, such that an array of related coronaviruses needed to be compared to the query sequences (i.e., SARS-like virus CX1 or SARS-like virus LS1). We have updated this figure as suggested by Reviewer 3 to set SARS-like virus CX1 and the SARS-CoV-2 reference sequences as the queries. The plot for SARS-like virus LS1 was moved to the supplementary materials (because it did not show any recombination). A new plot for SARS-CoV-2 has been included to show the potential origin of the early genome sequence of SARS-CoV-2. Finally, we have now used consistent colours among plots as requested.

Reviewer #2 (Remarks to the Author):

The manuscript by Jing Wang et al. presented an interesting approach to look at virome at individual bat level. The result is very interesting showing a large diversity of viruses carried by different bat species and a frequent virus sharing among bat species. Interestingly, the authors discovered a recombinant SARS-CoV- like virus with evidence of history of recombination with ability to utilize human ACE2 receptor for cell entry. The manuscript is well thought and written. There are some small points to be clarified.

- What does BtSY stand for? BtSY seems to be associated to different viruses. Using BtSY1 and BtSY2 may not be appropriate to represent SARS-like virus. For example, line 136 has Bat RVJ-like rotavirus BtSY1.

Response: We renamed all viruses based on viral taxonomy and the place where it was detected. For example, Bat SARS-like virus BtSY1 is now renamed to Bat SARS-like coronavirus LS1, which means that it is a SARS-related coronavirus that was detected in Lushui country. We provide a table of all old names and corresponding new names (please refer to name_mapping.xlsx). Also, we refer to the two SARS-like viruses in the main text by the abbreviated names “SARS-like virus CX1” and “SARS-like virus LS1”, to eliminate ambiguity.

- Lines 154, though we can find the information in the figure 4, it would nice to give some context of the detection of the two SARS-like virus in bats. What were the bat species? Were the virus detected in multiple bats?

Response: Both SARS-like viruses were detected in multiple bat species. Bat SARS-like coronavirus CX1 was detected in *Rh. pusillus* and *Rh. marshalli*, and Bat SARS-like coronavirus LS1 was detected in *Rh. macrotis* and *Rh. thomasi*. We add a new Table 1 to the main text to show where the five viruses of concern were detected and their prevalence.

- Rotavirus seems to be predominantly detected in different bat genus. Do authors have any

explanation? It would be interesting to develop a bit more in discussion. Rotavirus A is the most common species to cause infection in human. How close the virus detected in bats to the ones that cause disease in human?

Response: We added some more discussion on this topic (lines 655-659). However, the analysis of virus host range is inevitably influenced by uneven sampling efforts, so we do not want to overly extend the discussion of host range to avoid over-interpretation. We have also added a Limitations section (lines 664-674) to inform the readers of this issue. Finally, we have added a new Table 1 to show the similarity of viruses of concern to known human and livestock pathogens.

- Line 264, Sample collection. Were the capture done in open air or at the entrance of cave? Do the locations present any interfaces with human or other animals? How the locations were selected?

Response: The selection of bat sampling sites includes factors such as whether there are caves or other environments where bats are known to inhabit, the possibility of contact with humans, the presence of unexplained fever patients, and border areas. Specifically, bats were mainly collected from mountain caves (in Baoshan, Chuxiong, and Mengla), orchards (in Wanding), and crevices in the wild cliffs (in Lushui and Zhenkang). Bats collected from orchards have close contact with humans, those collected from mountain caves have occasional contact with a small number of people and animals, while those collected from crevices in the wild cliffs have minimal contact with humans. We added some more details about sampling to the Methods section (lines 700-710).

- Although the *in silico* tools used to evaluate human ACE2 binding potential of BtSY2 gave strong evidence showing BtSY2 may utilize human ACE2 receptor for cell entry, I wonder why authors did not perform *in vitro* experiment which is pretty common in the first place.

Response: We synthesized the RBD of SARS-like virus CX1 (original "BtSY2"), and performed two *in vitro* assays, namely the ELISA and BLI assays, to quantify the binding affinity to hACE2 (see Figure 7). Both results suggested that RBD of SARS-like virus CX1 can indeed bind to the hACE2 receptor, although with lower affinity than either SARS-CoV-2 or SARS-CoV. The corresponding methods are described in lines 868-889 and the results in lines 547-554.

Reviewer #3 (Remarks to the Author):

The work presented by Wang et al. described the mammalian virome of individual bat rectal swabs collected over a 4-year period in Yunnan, China. The findings of the study are not very innovative compared to other studies, but sarbecoviruses described here are of interest, especially regarding the origin of SARS-CoV-1 and SARS-CoV-2 viruses.

Major comments:

1. The mention "likely to be pathogenic for humans and livestock" (regarding coronaviruses) present in the abstract and through the whole manuscript is a bit speculative and need to be mitigated. The conservation of contact RBD/ACE2 residues and MD simulations give some

evidence of a possible human infection but do not inform on the putative pathogenicity of the virus for humans or livestock.

Response: We agree with the reviewer on this point. We have therefore toned down the claims about putative pathogenicity in the revised manuscript (e.g., lines 622-624) and added a new Limitation section (lines 664-674). We also performed two *in vitro* assays that provided additional evidence that Bat SARS-like coronavirus CX1 (original “BtSY2”) may be able to utilize the hACE2 receptor.

2. Co-infection is a pre-requisite for virus recombination, and I agree with the authors that the study of individual virome is important to study co-infection. However, recombination can occur only when a given cell is concomitantly infected by at least two viruses; and the description of viral communities of a bat rectal swab does not inform on which cell is infected. So I strongly suggest the authors to add a paragraph to discuss this issue and to change the titer because their study does not provide rational evidence of co-infection.

Response: We thank the reviewer for pointing out this. We recognize that “co-infection” typically refers to the simultaneous infection of cells by virologists, although it is also used to describe the simultaneous infection of the same host individual by ecologists (e.g., Perkins and Rohr 2020). We added a new “Limitations” section (lines 664-674) to clarify the difference between co-infection of a host and co-infection of a cell.

Perkins, A. T., & Rohr, J. R. (2020). Theories of diversity in disease ecology. *Theoretical Ecology: Concepts and Applications*; eds. Kevin McCann and Gabriel Gellner, Oxford University Press, London.

3. Regarding the cross-species transmission of viruses among bats, an important information has to be added in the manuscript: when the authors claim that the same virus species is present in different bat species, did they perform a distance matrix (or another type of analysis) to compare the genetic distance/identity between the different strains of a given virus species identified in the different bats? For example, is it the same BtSY2 coronavirus (and how close are these strains) that is shared between *R. pusillus* and *R. marshalli*? Such analysis will give additional evidence of the cross-species transmission of the same virus to different bat species. And if it the case, how do the authors explain this cross-species transmission: co-living in the same caves? Another explanation?

Response: We did perform multiple sequence alignments and calculated the percentage nucleotide identity for each virus. This result is presented briefly in the main text (lines 219-221) and a figure (new Fig.S4) of pairwise distances between viruses has been added to the supplementary materials. The two variants of SARS-like coronavirus CX1 are highly similar (98.6% whole-genome nucleotide identity), and the two positive bat individuals (one *R. pusillus* and one *R. marshalli*) were captured from the same cave and at the same time. We have added an explanation to this in the Discussion (lines 590-599). Briefly, as the two host species are closely related it is possible that the virus might be able to infect both species without many adaptations. In addition, as the two bat species co-habit the same cave, there is a strong chance that they will interact, in turn providing evidence for virus transmission.

4. Following this point of cross-species transmission of viruses among bats, I am not sure that the network analysis presented in Figure 4 is suitable and sufficient to claim that “more closely phylogenetically related or closely geographically located bat individuals had more similar virome compositions and had more virus species in common” since the analysis was restricted to mammalian-infecting viruses. Could the authors perform a more comprehensive description of the individual virome that include viruses associated to prokaryotes, plants, etc.? It will help to distinguish the core virome (shared by different bat species/genera) and the virome which is specific to a given bat species or genus.

Response: We now present the results of total virome in supplementary materials. Partial Mantel tests revealed that our observation that “more closely phylogenetically related or closely geographically located bat individuals had more similar virome compositions and had more virus species in common” is also true for the total RNA virome (Table. S5, Figure. S5-7). However, the total virome is very complex, some viruses are associated with gut microbiome and some are associated with plants and invertebrates (probably from the diet). Although we agree that some of these viruses may interact with bat-associated viruses or even impact bat health, the specific biological importance of these viruses is unclear. Therefore, we wish to focus on the bat-associated virome in this paper.

5. The results regarding the BtSY2 coronavirus are interesting, but a detailed recombination analysis is needed (with for example GARD or 3SEQ). SimPlot is not a tool to determine recombinant breakpoints. How do the authors determine the coordinates of breakpoint mentioned line 193 page 7? In addition, the MD simulation and the analysis of mismatches in the ACE2/RBD contact residues are not sufficient to claim that “these data tentatively suggest that BtSY2 may be able to replicate rapidly with similar virulence as SARS-CoV”. It is too speculative.

Response: We performed recombination analysis and determined break points using 3SEQ. Furthermore, we agree with the reviewer that *in silico* and *in vitro* assays of receptor binding are not sufficient to infer pathogenicity, and have therefore removed the claims about the pathogenicity of SARS-like virus CX1. And added a Limitation section to clarify this (lines 664-674).

Other comments:

1. A table (or the update of Table S5) should be added to present the horizontal genome coverage achieved for the viruses of concern, with the description of the closest viral sequence and the % identity in the different genes. In fact, we do not know in the manuscript if the authors obtained the full genomes for these viruses of concern.

Response: We obtained full genomes for four of the five viruses of concern (the exception was Bat HKU2-like coronavirus LS1, 3087 bp). To confirm the genome, we mapped the reads from library S18LSBatR79, where HKU2-like virus LS1 was detected, to the reference genome of *Rhinolophus* bat coronavirus HKU2 (NC_009988.1). The mapping result reveals 28.9% percent genome coverage and is presented in Fig. S11. We added new Table 1 and Table S3 to show the completeness, read depth and coverage of all viral genomes found in this study.

2. Please add a conclusion of the study.

Response: We added a conclusion to the end of discussion (lines 675-683), to read

“In conclusion, using an individual virome approach we revealed a high frequency of virus co-infection and inter-species transmission among bats. The discovery of a diverse array of bat-associated viruses, including those potentially pathogenic to humans and livestock, emphasizes the need for continued vigilance in monitoring bat populations as potential sources of emerging infectious diseases. The identification of a novel recombinant SARS-like coronavirus that can utilize the human ACE2 receptor raises concerns about the potential for future zoonotic spillover events. Further research into the diversity and abundance of viruses within bats is necessary to better understand the risks associated with zoonotic transmission and to inform the development of strategies for disease prevention and control”.

3. Do the authors have studied the effect of season or gestation stage of bats o sample location on the composition of viromes? These extrinsic factors may play an important role in the difference observed in virus abundance and diversity.

Response: We considered the effect of season. The sampling was done either during summer (July-August) and winter (November-December), and this effect was removed in analyses of virome composition by including the date of sampling in statistical models. As for the gestation stage issue, we did not collect bats in pregnancy.

Minor comments:

1. Introduction line 86 page 4: “A number of highly pathogenic viruses have been detected there”: give some example, because bat-borne relatives of SARS-CoV-1 and SARS-CoV-2 are not “highly pathogenic”

Response: We reworded this sentence: “a number of highly pathogenic viruses” is changed to “a number of potential zoonotic viruses”

2. Results line 120 page 5: “We consider those with relatively high viral load (reads per million total reads > 1) as true positives”. How do you defined this threshold?

Response: This threshold level has been validated in our previous publications. For example, in Shi et al. 2022 we used RT-PCR to confirm the meta-transcriptomics results and found that the RPM>1 removes most false positives. In this study, we also validated a subset of viruses, which meets RPM>0 but have low read coverage (<30%), using RT-PCR to remove false positives.

Shi, M., Zhao, S., Yu, B., Wu, W. C., Hu, Y., Tian, J. H., ... & Zhang, Y. Z. (2022). Total infectome characterization of respiratory infections in pre-COVID-19 Wuhan, China. *PLoS Pathogens*, 18(2), e1010259.

3. Line 148 page 6: typo “what” instead of “that”

Response: Revised accordingly.

4. Line 158 page 6: NTD and RBD are not genes but functional domains of the spike

Response: Revised accordingly.

5. Line 167 page 6 “The nucleotide identity between their RdRp was less than 80%, so we demarcated them as three different types”. Please use the ICTV demarcation criteria.

Response: We have now applied the suggested ICTV criteria (Matthijnssen *et al.* 2008, 2011) to demarcate the genotypes of Rotavirus A. Based on the ICTV criteria, the three “types” of Rotavirus A that we have identified indeed belong to three different genotypes. The standard genotype names of Bat Rotavirus A CX1, WD1 and WD2 are G3-P[10]-I17-R8-C12-M11-Ax-N18-T14-E13-H7, Gx-P[12]-I25-R19-Cx-M18-Ax-N19-T20-Ex-H20, and Gx-P[48]-I25-R19-C18-M18-Ax-N19-T20-E25-H20 respectively, in which “x” represent unclassified genotypes (no matches to existing genotypes in database). Notably, Rotavirus A is a phylogenetically divergent group of viruses that forms several clades associated with specific host taxa, and many of these are unable to infect humans. Therefore, we distinguished different genotypes of Rotavirus A rather than treating them as a single genotype.

Matthijnssens, J., Ciarlet, M., Rahman, M., Attoui, H., Bányai, K., Estes, M. K., ... & Van Ranst, M. (2008). Recommendations for the classification of group A rotaviruses using all 11 genomic RNA segments. *Archives of virology*, 153, 1621-1629.

Matthijnssens, J., Ciarlet, M., McDonald, S. M., Attoui, H., Bányai, K., Brister, J. R., ... & Van Ranst, M. (2011). Uniformity of rotavirus strain nomenclature proposed by the Rotavirus Classification Working Group (RCWG). *Archives of virology*, 156, 1397-1413.

6. Line 241 page 8 “and even has slightly higher affinity than SARS-CoV-2 Wuhan-Hu-1”: how do the authors claim that?

Response: We removed this claim from the text. The newly added *in vitro* assays now suggest that the hACE2-binding affinity of SARS-like virus CX1 is weaker than SARS-CoV-2.

7. Figure 6B: for a better comprehension, I would rather present SARS-CoV-2 Wuhan as a reference for Simplot analysis at the whole genome level. And please use the same strains for Simplot analyses at the whole genome and at the spike levels: reader may not know all the strains used here.

Response: We agree with the reviewer and have made revisions to Figure 6 accordingly.

8. Figure 6B, legend: “Recombination analysis”. NO, SimPlot is not a recombination analysis tool.

Response: We now performed recombination analysis using 3SEQ.

9. Table S3 is not referred in the text

Response: The original Table S3 presented raw data on viral abundance. This has now been provided as Supplemental Data. And we added a new Table S3 showing genome coverage.

REVIEWER COMMENTS

Reviewer #1 (Remarks to the Author):

I think the authors have done a nice job addressing my concerns and the points brought up by the other reviewers, especially with regards to toning down some claims and adding a clear limitations section to the manuscript. The only additional place I would suggest the authors tone down their language is in the abstract, specifically on Line 59, where the authors state: “which in turn facilitates virus recombination and reassortment.” As pointed out by Reviewer #3, co-infection in the same organism doesn’t always lead to recombination, so it may be helpful to clarify that this “CAN facilitate virus recombination and reassortment.”

My only other comment is to request a little bit more clarification on the methods, as I’m still not completely convinced that potential contamination and other confounders are completely accounted for (though it’s possible they are, and this information simply needs to be more clearly conveyed). Specifically, I noted in Supplementary Table 2 that there are a number of viruses with relatively low species identity. I imagine this is fine for the family/genus level analyses described around Line 161, but what about when looking at specific species that are shared? I am also a little confused about the claim, “The whole genome nucleotide identity of the sequences from each viral species in different bat species ranged from 78% to 100%, with most having nucleotide identities >90%” since this isn’t what I am seeing in Supplementary Table 3. Perhaps I am not understanding which tables correspond to which analyses, so if the authors could clarify this that would be helpful (putting these tables in an Excel spreadsheet instead where all columns can be shown for all samples at once would certainly help).

Relatedly, I’m not sure the authors completed addressed my question about negative controls or other contamination controls (only positive controls were mentioned). What did the authors do to ensure sequences found in multiple bat samples were not due to contamination during the sequencing process?

Thank you to the authors for addressing these concerns and making the underlying data a bit more accessible. Once they do this, I would be happy to see this manuscript published.

Reviewer #2 (Remarks to the Author):

I am happy with the authors' reply and the edit to the manuscript.

Reviewer #3 (Remarks to the Author):

I would like to thank the authors for their answers and their additional experiments and analyses, which I think add value to the manuscript.

I have one remaining concern, regarding the Main Comment no 3.

The two variants of BsSY2 sarbecoviruses present a whole genome identity of 98.6%, which classify them as two different strains of the same virus species (as noted by the authors).

Since these variants were found in two different bat species (Rh. marshalli = published / Rh. pusillus = unpublished), I strongly encourage the authors to make available the complete genome of these two variants (in GenBank for example), even if they were sampled at the same time and in the same cave, because each novel complete genome of sarbecoviruses is important for the scientific community.

The same comment is applicable for BtSY1 sarbecovirus found in Rh. thomasi (published) and Rh. macrotis (unpublished). How close are the two strains?

Reviewer #1 (Remarks to the Author):

I think the authors have done a nice job addressing my concerns and the points brought up by the other reviewers, especially with regards to toning down some claims and adding a clear limitations section to the manuscript. The only additional place I would suggest the authors tone down their language is in the abstract, specifically on Line 59, where the authors state: "which in turn facilitates virus recombination and reassortment." As pointed out by Reviewer #3, co-infection in the same organism doesn't always lead to recombination, so it may be helpful to clarify that this "CAN facilitate virus recombination and reassortment."

Response: Thank you very much for your diligent effort in reviewing our manuscript and providing helpful advices. We have now reworded "can facilitate" to "may facilitate" in the Abstract.

My only other comment is to request a little bit more clarification on the methods, as I'm still not completely convinced that potential contamination and other confounders are completely accounted for (though it's possible they are, and this information simply needs to be more clearly conveyed). Specifically, I noted in Supplementary Table 2 that there are a number of viruses with relatively low species identity. I imagine this is fine for the family/genus level analyses described around Line 161, but what about when looking at specific species that are shared? I am also a little confused about the claim, "The whole genome nucleotide identity of the sequences from each viral species in different bat species ranged from 78% to 100%, with most having nucleotide identities >90%" since this isn't what I am seeing in Supplementary Table 3. Perhaps I am not understanding which tables correspond to which analyses, so if the authors could clarify this that would be helpful (putting these tables in an Excel spreadsheet instead where all columns can be shown for all samples at once would certainly help).

Response: We are sorry for the confusion. The "Supplementary Table S2" described the protein identity of the viruses we found to existing records in a database (i.e., the NCBI non-redundant protein (nr) database). This described the relation of the viruses we found to known viruses (i.e., the existing records in database). Many of the viruses we found indeed had low similarity to known viruses, and we designated them as novel viral species according to the International Committee on Taxonomy of Viruses (ICTV) criteria. This allowed us to perform analysis on both viral family and species level.

By saying "The whole genome nucleotide identity of the sequences from each viral species in different bat species ranged from 78% to 100%, with most having nucleotide identities >90%", the intended point is that the genome

sequences of different strains of the same viral species detected in this study were highly similar. This addressed the concern of reviewer #3 (previous major comment 3), and such finding was supported by Supplementary Fig.S4, where we compared the genome identity of each strain of each virus species to its representative genome (we used the genome sequence of one strain per species as representative). These "strains" and "representative genomes" were assembled from our samples, instead of being retrieved from database.

Supplementary Table S3 described the genome coverage and read depth of each strain. In this revision, we uploaded these tables in the format of Excel spreadsheet as "Supplementary Files", so that readers can more easily evaluate these data.

Relatedly, I'm not sure the authors completely addressed my question about negative controls or other contamination controls (only positive controls were mentioned). What did the authors do to ensure sequences found in multiple bat samples were not due to contamination during the sequencing process?

Response: We apologize for the confusion again. These measures of false-positive control were conducted exactly for the purpose of reducing artefacts due to contamination during the sequencing process. We have now made this explicit in the *Methods* section (lines 404-409). Below we explained how these false-positive controls can reduce the chance that sequences found in multiple bat samples were not due to contamination during the sequencing process.

First, index-hopping is a very common reason that may generate artefact sequences. It is a phenomenon that occurs during high-throughput sequencing, when reads from one sample being erroneously assigned to another sample. This phenomenon typically happens when the read abundance is high in one sample, so that these reads may leak into other samples in the same sequencing chip by error. Therefore, we minimized the effect of index-hopping by applying a read abundance threshold relative to the maximal read abundance within each sequencing lane (described in detail between line 401-406).

Second, our previous studies suggested that viruses at low abundance are likely to be artefacts produced by sequencing and downstream bioinformatics analysis. And thus, we applied an RPM>1 filter to reduce these artefacts (lines 398-401).

Third, to further ensure sequences found in multiple bat samples were not due to contamination during the sequencing process, we complemented the high-throughput sequencing results with reverse-transcription PCR (RT-PCR) method (line 407). We specifically confirmed the existence of the five viruses of

concerns and those viruses with low genome coverage (<30%, see Supplementary Table S3) using RT-PCR. The double-confirmation can largely reduce the probability that sequences found in multiple bat samples were due to contamination during the sequencing process.

Thank you to the authors for addressing these concerns and making the underlying data a bit more accessible. Once they do this, I would be happy to see this manuscript published.

Reviewer #2 (Remarks to the Author):

I am happy with the authors' reply and the edit to the manuscript.

Response: We greatly appreciate your review of our manuscript. Thank you very much.

Reviewer #3 (Remarks to the Author):

I would like to thank the authors for their answers and their additional experiments and analyses, which I think add value to the manuscript.

I have one remaining concern, regarding the Main Comment no 3.

The two variants of BsSY2 sarbecoviruses present a whole genome identity of 98.6%, which classify them as two different strains of the same virus species (as noted by the authors).

Since these variants were found in two different bat species (Rh. marshalli = published / Rh. pusillus = unpublished), I strongly encourage the authors to make available the complete genome of these two variants (in GenBank for example), even if they were sampled at the same time and in the same cave, because each novel complete genome of sarbecoviruses is important for the scientific community.

The same comment is applicable for BtSY1 sarbecovirus found in Rh. thomasi (published) and Rh. macrotis (unpublished). How close are the two strains?

Response: We sincerely thank you for reviewing our manuscript. The three strains of SARS-like virus LS1 shared 99%~100% nucleotide identity with each other. We have uploaded all these genome sequences to GenBank now. It will take several weeks for NCBI to process the submitted sequences, so we temporarily provide access to these sequences in our GitHub repository (<https://github.com/Augustpan/Individual-Bat-Virome>).

REVIEWERS' COMMENTS

Reviewer #1 (Remarks to the Author):

I am happy with the authors' responses to my previous review. Thank you!

REVIEWERS' COMMENTS

Reviewer #1 (Remarks to the Author):

I am happy with the authors' responses to my previous review. Thank you!

AUTHORS' RESPONSE

Thank you very much for reviewing our manuscript!